# (Almost) Provable Error Bounds Under Distribution Shift via Disagreement Discrepancy

**Elan Rosenfeld**
Machine Learning Department
Carnegie Mellon University
elan@cmu.edu

**Saurabh Garg**
Machine Learning Department
Carnegie Mellon University

## Abstract

We derive a new, (almost) guaranteed upper bound on the error of deep neural networks under distribution shift using unlabeled test data. Prior methods are either vacuous in practice or accurate on average but heavily underestimate error for a sizeable fraction of shifts. In particular, the latter only give guarantees based on complex continuous measures such as test calibration, which cannot be identified without labels, and are therefore unreliable. Instead, our bound requires a simple, intuitive condition which is well justified by prior empirical works and holds in practice effectively 100% of the time. The bound is inspired by $\mathcal{H}\Delta\mathcal{H}$-divergence but is easier to evaluate and substantially tighter, consistently providing non-vacuous test error upper bounds. Estimating the bound requires optimizing one multiclass classifier to disagree with another, for which some prior works have used sub-optimal proxy losses; we devise a "disagreement loss" which is theoretically justified and performs better in practice. We expect this loss can serve as a drop-in replacement for future methods which require maximizing multiclass disagreement. Across a wide range of natural and synthetic distribution shift benchmarks, our method gives valid error bounds while achieving average accuracy comparable to—though not better than—competitive estimation baselines.

## 1 Introduction

When deploying a model, it is important to be confident in how it will perform under inevitable distribution shift. Standard methods for achieving this include data dependent uniform convergence bounds [41, 6] (typically vacuous in practice) or assuming a precise model of how the distribution can shift [57, 10]. Unfortunately, it is difficult or impossible to determine how severely these assumptions are violated by real data [62], so practitioners usually cannot trust such bounds with confidence.

To better estimate test performance in the wild, some recent work instead tries to directly predict accuracy of neural networks using unlabeled data from the test distribution of interest [21, 3, 40]. While these methods predict the test performance surprisingly well, they lack pointwise trustworthiness and verifiability: their estimates are good on average over all distribution shifts, but they provide no signal of the quality of any individual prediction (here, each point is a single test *distribution*, for which a method predicts a classifier's average accuracy). Because of the opaque conditions under which these methods work, it is also difficult to anticipate their failure cases—indeed, it is reasonably common for them to substantially overestimate test accuracy for a particular shift, which is problematic when optimistic deployment would be very costly. Worse yet, we find that this gap *grows with test error* (Figure 1), making these predictions least reliable precisely when their reliability is most important. **Although it is clearly impossible to guarantee upper bounds on test error for all shifts,** there is still potential for error bounds that are intuitive and reasonably trustworthy.

37th Conference on Neural Information Processing Systems (NeurIPS 2023).

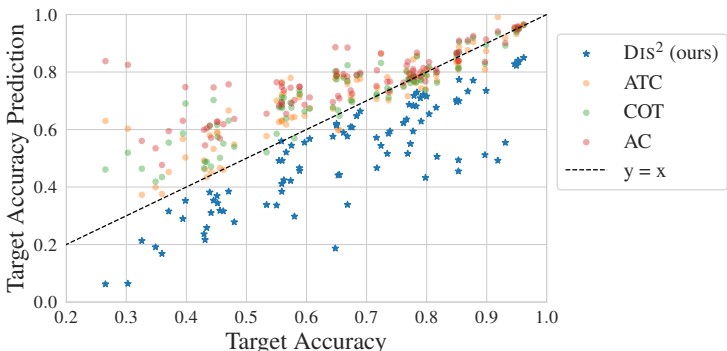

Figure 1: **Our bound vs. three prior methods for estimation across a wide variety of distribution shift benchmarks (e.g., WILDs, BREEDs, DomainNet) and training methods (e.g., ERM, FixMatch, BN-adapt).** Prior methods are accurate on average, but it is difficult or impossible to know when a given prediction is reliable and why. Worse yet, they usually overestimate accuracy, with the gap growing as test accuracy decreases—*this is precisely when a reliable, conservative estimate is most desirable.* Instead, DIS$^2$ maximizes the **dis**agreement **dis**crepancy to give a reliable error upper bound which holds effectively 100% of the time. See Appendix F for stratification by training method.

In this work, we develop a method for (almost) provably bounding test error of classifiers under distribution shift using unlabeled test points. Our bound's only requirement is a simple, intuitive, condition which describes the ability of a hypothesis class to achieve small loss on a particular objective defined over the (unlabeled) train and test distributions. Inspired by $\mathcal{H}\Delta\mathcal{H}$-divergence [41, 7], our method requires training a critic to maximize agreement with the classifier of interest on the source distribution while simultaneously maximizing *disagreement* on the target distribution; we refer to this joint objective as the *Disagreement Discrepancy*, and so we name the method DIS$^2$. We optimize this discrepancy over linear classifiers using deep features—or linear functions thereof—finetuned on only the training set. Recent evidence suggests that such representations are sufficient for highly expressive classifiers even under large distribution shift [61]. Experimentally, we find that our bound is valid effectively 100% of the time,[1] consistently giving non-trivial lower bounds on test accuracy which are reasonably comparable to competitive baselines.

Additionally, our proof of the bound leads to a natural (post-hoc) hypothesis test of the validity of its lone assumption. This provides an unusually strong positive signal: for more than half the datasets we evaluate we *prove* with high probability that the assumption holds; the corresponding indicator that it does not hold never occurs. We also show that it is possible to approximately test this bound's likelihood of being valid *a priori* using only unlabeled data: the optimization process itself provides useful information about the bound's validity, and we use this to construct a score which linearly correlates with the tightness of the bound. This score can then be used to relax the original bound into a sequence of successively tighter-yet-less-conservative estimates, interpolating between robustness and accuracy and allowing a user to make estimates according to their specific risk tolerance.

While maximizing agreement is statistically well understood, our method also calls for maximizing *dis*agreement on the target distribution. This is not so straightforward in the multiclass setting, and we observe that prior works use unsuitable losses which do not correspond to minimizing the 0-1 loss of interest and are non-convex (or even concave) in the model logits [12, 50, 23]. To rectify this, we derive a new "disagreement loss" which serves as an effective proxy loss for maximizing multiclass disagreement. Experimentally, we find that minimizing this loss results in lower risk (that is, higher disagreement) compared to prior methods, and we believe it can serve as a useful drop-in replacement for any future methods which require maximizing multiclass disagreement.

Experiments across numerous vision datasets (BREEDs [65], FMoW-WILDs [35], Visda [51], Domainnet [53], CIFAR10, CIFAR100 [36] and OfficeHome [69]) demonstrate the effectiveness of our bound. Though DIS$^2$ is competitive with prior methods for error estimation, **we emphasize that our focus is *not* on improving raw predictive accuracy**—rather, we hope to obtain reliable (i.e.,

---

[1]The few violations are expected a priori, have an obvious explanation, and only occur for a specific type of learned representation. We defer a more detailed discussion of this until after we present the bound.

correct), reasonably tight bounds on the test error of a given classifier under distribution shift. In particular, while existing methods tend to severely overestimate accuracy as the true accuracy drops, our bound maintains its validity while remaining non-vacuous, *even for drops in accuracy as large as 70%*. In addition to source-only training, we experiment with unsupervised domain adaptation methods that use unlabeled target data and show that our observations continue to hold.

## 2   Related Work

**Estimating test error with unlabeled data.**   The generalization capabilities of overparameterized models on in-distribution data have been extensively studied using conventional machine learning tools [46, 47, 45, 48, 17, 5, 73, 39, 42]. This research aims to bound the generalization gap by evaluating complexity measures of the trained model. However, these bounds tend to be numerically loose compared to actual generalization error [70, 43]. Another line of work instead explores the use of unlabeled data for predicting in-distribution generalization [56, 55, 20, 44, 32]. More relevant to our work, there are several methods that predict the error of a classifier under distribution shift with unlabeled test data: (i) methods that explicitly predict the correctness of the model on individual unlabeled points [14, 15, 8]; and (ii) methods that directly estimate the overall error without making a pointwise prediction [9, 24, 12, 21, 3].

To achieve a consistent estimate of the target accuracy, several works require calibration on the target domain [32, 24]. However, these methods often yield poor estimates because deep models trained and calibrated on a source domain are not typically calibrated on previously unseen domains [49]. Additionally, [14, 24] require a subset of labeled target domains to learn a regression function that predicts model performance—but thus requires significant a priori knowledge about the nature of shift that, in practice, might not be available before models are deployed in the wild.

Closest to our work is [12], where the authors use domain-invariant predictors as a proxy for unknown target labels. However, like other works, their method only *estimates* the target accuracy—the actual error bounds they derive are not computable in practice. Second, even their estimate is computationally demanding and relies on multiple approximations, tuning of numerous hyperparameters, e.g. lagrangian multipliers; as a result, proper tuning is difficult and the method does not scale to modern deep networks. Finally, they suggest minimizing the (concave) negative cross-entropy loss, but we show that this can be a poor proxy for maximizing disagreement, instead proposing a more suitable replacement which we find performs much better.

**Uniform convergence bounds.**   Our bound is inspired by classic analyses using $\mathcal{H}$- and $\mathcal{H}\Delta\mathcal{H}$-divergence [41, 6, 7]. These provide error bounds via a complexity measure that is both data- and hypothesis-class-dependent. This motivated a long line of work on training classifiers with small corresponding complexity, such as restricting classifiers' discriminative power between source and target data [18, 67, 38, 72]. Unfortunately, such bounds are often intractable to evaluate and are usually vacuous in real world settings. We provide a more detailed comparison to our approach in Section 3.1.

## 3   Deriving an (Almost) Provable Error Bound

**Notation.**   Let $\mathcal{S}, \mathcal{T}$ denote the source and target (train and test) distributions, respectively, over labeled inputs $(x, y) \in \mathcal{X} \times \mathcal{Y}$, and let $\hat{\mathcal{S}}, \hat{\mathcal{T}}$ denote sets of samples from them with cardinalities $n_S$ and $n_T$ (they also denote the corresponding empirical distributions). Recall that we observe only the covariates $x$ without the label $y$ when a sample is drawn from $\mathcal{T}$. We consider classifiers $h : \mathcal{X} \to \mathbb{R}^{|\mathcal{Y}|}$ which output a vector of logits, and we let $\hat{h}$ denote the particular classifier whose error we aim to bound. Generally, we use $\mathcal{H}$ to denote a hypothesis class of such classifiers. Where clear from context, we use $h(x)$ to refer to the argmax logit, i.e. the predicted class. We treat these classifiers as deterministic throughout, though our analysis can easily be extended to probabilistic classifiers and labels. For a distribution $\mathcal{D}$ on $\mathcal{X} \times \mathcal{Y}$, let $\epsilon_{\mathcal{D}}(h, h') := \mathbb{E}_{\mathcal{D}}[\mathbf{1}\{\arg\max_y h(x)_y \neq \arg\max_y h'(x)_y\}]$ denote the one-hot disagreement between classifiers $h$ and $h'$ on $\mathcal{D}$. Let $y^*$ represent the true labeling function such that $y^*(x) = y$ for all samples $(x, y)$; with some abuse of notation, we write $\epsilon_{\mathcal{D}}(h)$ to mean $\epsilon_{\mathcal{D}}(h, y^*)$, i.e. the 0-1 error of classifier $h$ on distribution $\mathcal{D}$.

The bound we derive in this work is extremely simple and relies on one new concept:

**Definition 3.1.** The *disagreement discrepancy* $\Delta(h, h')$ is the disagreement between $h$ and $h'$ on $\mathcal{T}$ minus their disagreement on $\mathcal{S}$:

$$\Delta(h, h') := \epsilon_\mathcal{T}(h, h') - \epsilon_\mathcal{S}(h, h').$$

We leave the dependence on $\mathcal{S}, \mathcal{T}$ implicit. Note that this term is symmetric in its arguments and signed—it can be negative. With this definition, we now have the following lemma:

**Lemma 3.2.** *For any classifier $h$, $\epsilon_\mathcal{T}(h) = \epsilon_\mathcal{S}(h) + \Delta(h, y^*)$.*

*Proof.* By definition, $\epsilon_\mathcal{T}(h) = \epsilon_\mathcal{S}(h) + (\epsilon_\mathcal{T}(h) - \epsilon_\mathcal{S}(h)) = \epsilon_\mathcal{S}(h) + \Delta(h, y^*)$. $\qquad\square$

We cannot directly use Lemma 3.2 to estimate $\epsilon_\mathcal{T}(\hat{h})$ because the second term is unknown. However, observe that $y^*$ is *fixed*. That is, while a learned $\hat{h}$ will depend on $y^*$ and $\mathcal{S}$—and therefore $\Delta(\hat{h}, y^*)$ may be large under large distribution shift—$y^*$ **is *not* chosen to maximize** $\Delta(\hat{h}, y^*)$ **in response to the $\hat{h}$ we have learned.** This means that for a sufficiently expressive hypothesis class $\mathcal{H}$, it should be possible to identify an alternative labeling function $h' \in \mathcal{H}$ for which $\Delta(\hat{h}, h') \geq \Delta(\hat{h}, y^*)$ (we refer to such $h'$ as the *critic*). In other words, we should be able to find an $h' \in \mathcal{H}$ which, *if it were the true labeling function*, would imply at least as large of a drop in accuracy from train to test as occurs in reality. This key observation serves as the basis for our bound, and we discuss it in greater detail in Section 3.1.

In this work we consider the class $\mathcal{H}$ of linear critics, with the features $\mathcal{X}$ defined as source-finetuned neural representations or the logits output by the classifier $\hat{h}$. Prior work provides strong evidence that this class has surprising capacity under distribution shift, including the possibility that functions very similar to $y^*$ lie in $\mathcal{H}$ [61, 34, 33]. We formalize this intuition with the following assumption:

**Assumption 3.3.** *Define $h^* := \arg\max_{h' \in \mathcal{H}} \Delta(\hat{h}, h')$. We assume*

$$\Delta(\hat{h}, y^*) \leq \Delta(\hat{h}, h^*).$$

Note that this statement is necessarily true whenever $y^* \in \mathcal{H}$; it only becomes meaningful when considering restricted $\mathcal{H}$, as we do here. Note also that this assumption is made specifically for $\hat{h}$, i.e. on a per-classifier basis. This is important because while the above may not hold for every classifier $\hat{h}$, it need only hold for the classifiers whose error we would hope to bound, which is in practice a very small subset of classifiers (such as those which can be found by approximately minimizing the empirical training risk via SGD). From Lemma 3.2, we immediately have the following result:

**Proposition 3.4.** *Under Assumption 3.3, $\epsilon_\mathcal{T}(\hat{h}) \leq \epsilon_\mathcal{S}(\hat{h}) + \Delta(\hat{h}, h^*)$.*

Unfortunately, identifying the optimal critic $h^*$ is intractable, meaning this bound is still not estimable—we present it as an intermediate result for clarity of presentation. To derive the practical bound we report in our experiments, we need one additional step. In Section 4, we derive a "disagreement loss" which we use to approximately maximize the empirical disagreement discrepancy $\hat{\Delta}(\hat{h}, \cdot) = \epsilon_{\hat{\mathcal{T}}}(\hat{h}, \cdot) - \epsilon_{\hat{\mathcal{S}}}(\hat{h}, \cdot)$. Relying on this loss, we instead make the assumption:

**Assumption 3.5.** *Suppose we identify the critic $h' \in \mathcal{H}$ which maximizes a concave surrogate to the empirical disagreement discrepancy. We assume $\Delta(\hat{h}, y^*) \leq \Delta(\hat{h}, h')$.*

This assumption is slightly stronger than Assumption 3.3—in particular, Assumption 3.3 implies with high probability a weaker version of Assumption 3.5 with additional terms that decrease with increasing sample size and a tighter proxy loss.[2] Thus, the difference in strength between these two assumptions shrinks as the number of available samples grows and as the quality of our surrogate objective improves. Ultimately, our bound holds without these terms, implying that the stronger assumption is reasonable in practice. We can now present our main result:

**Theorem 3.6** (Main Bound). *Under Assumption 3.5, with probability $\geq 1 - \delta$,*

$$\epsilon_\mathcal{T}(\hat{h}) \leq \epsilon_{\hat{\mathcal{S}}}(\hat{h}) + \hat{\Delta}(\hat{h}, h') + \sqrt{\frac{(n_S + 4n_T)\log 1/\delta}{2n_S n_T}}.$$

---

[2] Roughly, Assumption 3.3 implies $\Delta(\hat{h}, y^*) \leq \Delta(\hat{h}, h') + \mathcal{O}\left(\sqrt{\frac{\log 1/\delta}{\min(n_S, n_T)}}\right) + \gamma$, where $\gamma$ is a data-dependent measure of how tightly the surrogate loss bounds the 0-1 loss in expectation.

*Proof.* Assumption 3.5 implies $\epsilon_{\mathcal{T}}(\hat{h}) \leq \epsilon_{\mathcal{S}}(\hat{h}) + \Delta(\hat{h}, h') = \epsilon_{\mathcal{S}}(\hat{h}, y^*) + \epsilon_{\mathcal{T}}(\hat{h}, h') - \epsilon_{\mathcal{S}}(\hat{h}, h')$, so the problem reduces to upper bounding these three terms. We define the random variables

$$r_{\mathcal{S},i} = \begin{cases} 1/n_S, & h'(x_i) = \hat{h}(x_i) \neq y_i, \\ -1/n_S, & h'(x_i) \neq \hat{h}(x_i) = y_i, \\ 0, & \text{otherwise}, \end{cases} \qquad r_{\mathcal{T},i} = \frac{\mathbf{1}\{\hat{h}(x_i) \neq h'(x_i)\}}{n_T}$$

for source and target samples, respectively. By construction, the sum of all of these variables is precisely $\epsilon_{\hat{\mathcal{S}}}(\hat{h}, y^*) + \epsilon_{\hat{\mathcal{T}}}(\hat{h}, h') - \epsilon_{\hat{\mathcal{S}}}(\hat{h}, h')$ (note these are the empirical terms). Further, observe that

$$\mathbb{E}\left[\sum_{\hat{\mathcal{S}}} r_{\mathcal{S},i}\right] = \mathbb{E}_{\mathcal{S}}[\mathbf{1}\{\hat{h}(x_i) \neq y_i\} - \mathbf{1}\{\hat{h}(x_i) \neq h'(x_i)\}] = \epsilon_{\mathcal{S}}(\hat{h}, y^*) - \epsilon_{\mathcal{S}}(\hat{h}, h'),$$

$$\mathbb{E}\left[\sum_{\hat{\mathcal{T}}} r_{\mathcal{T},i}\right] = \mathbb{E}_{\mathcal{T}}[\mathbf{1}\{\hat{h}(x_i) \neq h'(x_i)\}] = \epsilon_{\mathcal{T}}(\hat{h}, h'),$$

and thus their expected sum is $\epsilon_{\mathcal{S}}(\hat{h}, y^*) + \epsilon_{\mathcal{T}}(\hat{h}, h') - \epsilon_{\mathcal{S}}(\hat{h}, h')$, which are the population terms we hope to bound. Now we apply Hoeffding's inequality: the probability that the expectation exceeds their sum by $t$ is no more than $\exp\left(-\frac{2t^2}{n_S(2/n_S)^2 + n_T(1/n_T)^2}\right)$. Solving for $t$ completes the proof. $\square$

*Remark* 3.7. While we state Theorem 3.6 as an implication, **Assumption 3.5 is *equivalent* to the stated bound up to finite-sample terms**. Our empirical findings (and prior work) suggest that Assumption 3.5 is reasonable in general, but this equivalence allows us to actually prove that it holds in practice for many shifts. We elaborate on this in Appendix E.

The core message behind Theorem 3.6 is that if there is a simple (i.e., linear) critic $h'$ with large disagreement discrepancy, the true $y^*$ could plausibly be this function, implying $\hat{h}$ could have high error—likewise, if no simple $y^*$ could hypothetically result in high error, we should expect low error.

*Remark* 3.8. Bounding error under distribution shift is fundamentally impossible without assumptions. Prior works which estimate accuracy using unlabeled data rely on experiments, suggesting that whatever condition allows their method to work holds in a variety of settings [21, 3, 40, 32, 24]; using these methods is equivalent to *implicitly* assuming that it will hold for future shifts. Understanding these conditions is thus crucial for assessing in a given scenario whether they can be expected to be satisfied.[3] It is therefore of great practical value that Assumption 3.5 is a simple, intuitive requirement: below we demonstrate that this simplicity allows us to identify potential failure cases *a priori*.

## 3.1 How Does $\mathrm{DIS}^2$ Improve over $\mathcal{H}$- and $\mathcal{H}\Delta\mathcal{H}$-Divergence?

To verifiably bound a classifier's error under distribution shift, one must develop a meaningful notion of distance between distributions. One early attempt at this was $\mathcal{H}$-*divergence* [6, 41] which measures the ability of a binary hypothesis class to discriminate between $\mathcal{S}$ and $\mathcal{T}$ in feature space. This was later refined to $\mathcal{H}\Delta\mathcal{H}$-*divergence* [7], which is equal to $\mathcal{H}$-divergence where the discriminator class comprises all exclusive-ors between pairs of functions from the original class $\mathcal{H}$. Though these measures can in principle provide non-vacuous bounds, they usually do not, and evaluating them is intractable because it requires maximizing an objective over all *pairs* of hypotheses. Furthermore, these bounds are overly conservative even for simple function classes and distribution shifts because they rely on uniform convergence. In practice, *we do not care* about bounding the error of all classifiers in $\mathcal{H}$—we only care to bound the error of $\hat{h}$. This is a clear advantage of $\mathrm{DIS}^2$ over $\mathcal{H}\Delta\mathcal{H}$.

**The true labeling function is never worst-case.**[4] More importantly, we observe that one should not expect the distribution shift to be *truly* worst case, because the test distribution $\mathcal{T}$ and ground truth

---

[3]Whether and when to trust a black-box estimate that is consistently accurate in all observed settings is a centuries-old philosophical problem [30] which we do not address here. Regardless, Figure 1 shows that these estimates are *not* consistently accurate, making interpretability that much more important.

[4]If it were, we'd see exactly 0% test accuracy—and when does that ever happen?

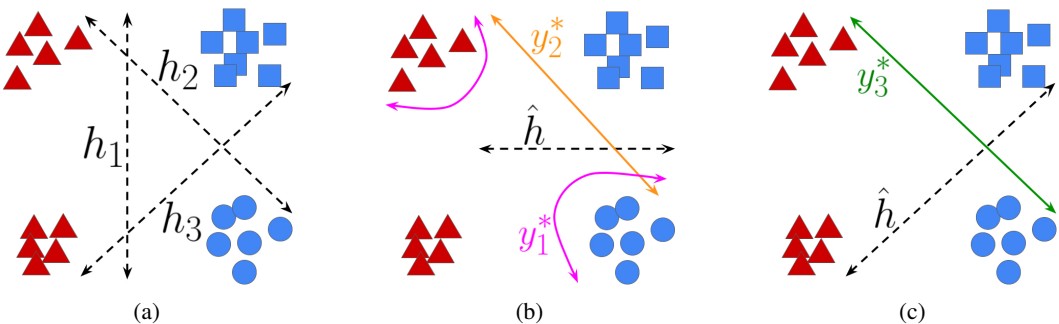

(a)                                (b)                                (c)

Figure 2: **The advantage of DIS$^2$ over bounds based on $\mathcal{H}$- and $\mathcal{H}\Delta\mathcal{H}$-divergence.** Consider the task of classifying circles and squares (triangles are unlabeled). **(a):** Because $h_1$ and $h_2 \oplus h_3$ perfectly discriminate between $\mathcal{S}$ (blue) and $\mathcal{T}$ (red), $\mathcal{H}$- and $\mathcal{H}\Delta\mathcal{H}$-divergence bounds are always vacuous. In contrast, DIS$^2$ is only vacuous when 0% accuracy is induced by a reasonably likely ground truth (such as $y_3^*$ in **(c)**, but not $y_1^*$ in **(b)**), and can often give non-vacuous bounds (such as $y_2^*$ in **(b)**).

$y^*$ are not chosen adversarially with respect to $\hat{h}$. Figure 2 gives a simple demonstration of this point. Consider the task of learning a linear classifier to discriminate between squares and circles on the source distribution $\mathcal{S}$ (blue) and then bounding the error of this classifier on the target distribution $\mathcal{T}$ (red), whose true labels are unknown and are therefore depicted as triangles. Figure 2(a) demonstrates that both $\mathcal{H}$- and $\mathcal{H}\Delta\mathcal{H}$-divergence achieve their maximal value of 1, because both $h_1$ and $h_2 \oplus h_3$ perfectly discriminate between $\mathcal{S}$ and $\mathcal{T}$. Thus both bounds would be vacuous.

Now, suppose we were to learn the max-margin $\hat{h}$ on the source distribution (Figure 2(b)). It is *possible* that the true labels are given by the worst-case boundary as depicted by $y_1^*$ (pink), thus "flipping" the labels and causing $\hat{h}$ to have 0 accuracy on $\mathcal{T}$. In this setting, a vacuous bound is correct. However, this seems rather unlikely to occur in practice—instead, recent experimental evidence [61, 34, 33] suggests that the true $y^*$ will be much simpler. The maximum disagreement discrepancy here would be approximately 0.5, giving a test accuracy lower bound of 0.5—this is consistent with plausible alternative labeling functions such as $y_2^*$ (orange). Even if $y^*$ is not linear, we still expect that *some* linear function will induce larger discrepancy; this is precisely Assumption 3.3. Now suppose instead we learn $\hat{h}$ as depicted in Figure 2(c). Then a simple ground truth such as $y_3^*$ (green) is plausible, which would mean $\hat{h}$ has 0 accuracy on $\mathcal{T}$. In this case, $y_3^*$ is also a critic with disagreement discrepancy equal to 1, and so DIS$^2$ would correctly output an error upper bound of 1.

**A setting where DIS$^2$ may be invalid.** There is one setting where it should be clear that Assumption 3.5 is less likely to be satisfied: when the representation we are using is explicitly regularized to keep $\max_{h' \in \mathcal{H}} \Delta(\hat{h}, h')$ small. This occurs for domain-adversarial representation learning methods such as DANN [18] and CDAN [38], which penalize the ability to discriminate between $\mathcal{S}$ and $\mathcal{T}$ in feature space. Given a critic $h'$ with large disagreement discrepancy, the discriminator $D(x) = \mathbf{1}\{\arg\max_y \hat{h}(x)_y = \arg\max_y h'(x)_y\}$ will achieve high accuracy on this task (precisely, $\frac{1+\Delta(\hat{h},h')}{2}$). By contrapositive, enforcing low discriminatory power means that the max discrepancy must also be small. It follows that for these methods DIS$^2$ should not be expected to hold universally, and in practice we see that this is the case (Figure 3). Nevertheless, when DIS$^2$ does overestimate accuracy, it does so by significantly less than prior methods.

## 4 Efficiently Maximizing the Disagreement Discrepancy

For a classifier $\hat{h}$, Theorem 3.6 clearly prescribes how to bound its test error: first, train a critic $h'$ on the chosen $\mathcal{X}$ to approximately maximize $\Delta(\hat{h}, h')$, then evaluate $\epsilon_{\hat{\mathcal{S}}}(\hat{h})$ and $\hat{\Delta}(\hat{h}, h')$ using a holdout set. The remaining difficulty is in identifying the maximizing $\hat{h}' \in \mathcal{H}$—that is, the one which minimizes $\epsilon_{\mathcal{S}}(\hat{h}, h')$ and maximizes $\epsilon_{\mathcal{T}}(\hat{h}, h')$. We can approximately minimize $\epsilon_{\mathcal{S}}(\hat{h}, h')$ by minimizing the sample average of the convex surrogate $\ell_{\text{logistic}} := -\frac{1}{\log |\mathcal{Y}|} \log \text{softmax}(h(x))_y$ as justified by statistical learning theory. However, it is less clear how to maximize $\epsilon_{\mathcal{T}}(\hat{h}, h')$.

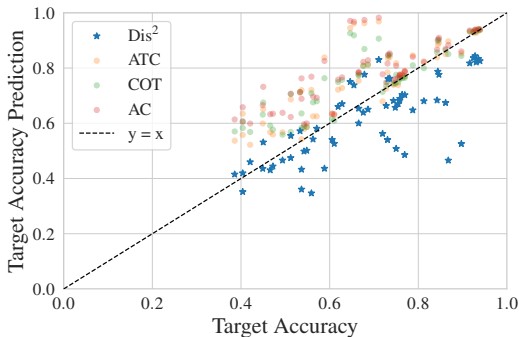

Figure 3: **D$\mathrm{IS}^2$ may be invalid when the features are regularized to violate Assumption 3.5.** Domain-adversarial representation learning algorithms such as DANN [18] and CDAN [38] indirectly minimize $\max_{h'\in\mathcal{H}} \Delta(\hat{h}, h')$, meaning the necessary condition is less likely to be satisfied. Nevertheless, when D$\mathrm{IS}^2$ does overestimate accuracy, it almost always does so by less than prior methods.

A few prior works suggest proxy losses for multiclass disagreement [12, 50, 23]. We observe that these losses are not theoretically justified, as they do not upper bound the 0-1 disagreement loss or otherwise do not meaningfully enforce that higher agreement causes higher loss. Furthermore, they are non-convex (or even concave) in the model logits, hindering optimization. Indeed, it is easy to identify simple settings in which minimizing these losses will result in a degenerate classifier with arbitrarily small loss but high agreement. Instead, we derive a new loss which satisfies the above desiderata and thus serves as a more principled approach to maximizing disagreement.

**Definition 4.1.** The *disagreement logistic loss* of a classifier $h$ on a labeled sample $(x, y)$ is defined as

$$\ell_{\mathrm{dis}}(h, x, y) := \frac{1}{\log 2} \log \left( 1 + \exp \left( h(x)_y - \frac{1}{|\mathcal{Y}| - 1} \sum_{\hat{y}\neq y} h(x)_{\hat{y}} \right) \right).$$

**Fact 4.2.** The disagreement logistic loss is convex in $h(x)$ and upper bounds the 0-1 disagreement loss (i.e., $\mathbf{1}\{\arg\max_{\hat{y}} h(x)_{\hat{y}} = y\}$). For binary classification, the disagreement logistic loss is equivalent to the logistic loss with the label flipped.

We expect that $\ell_{\mathrm{dis}}$ can serve as a useful drop-in replacement for any future algorithm which requires maximizing disagreement in a principled manner. We combine $\ell_{\mathrm{logistic}}$ and $\ell_{\mathrm{dis}}$ to arrive at the empirical disagreement discrepancy objective:

$$\hat{\mathcal{L}}_\Delta(h') := \frac{1}{|\hat{\mathcal{S}}|} \sum_{x\in\hat{\mathcal{S}}} \ell_{\mathrm{logistic}}(h', x, \hat{h}(x)) + \frac{1}{|\hat{\mathcal{T}}|} \sum_{x\in\hat{\mathcal{T}}} \ell_{\mathrm{dis}}(h', x, \hat{h}(x)).$$

By construction, $1 - \hat{\mathcal{L}}_\Delta(h')$ is concave and bounds $\hat{\Delta}(\hat{h}, h')$ from below. However, note that the representations are already optimized for accuracy on $\mathcal{S}$, which suggests that predictions will have low entropy and that the $1/\log |\mathcal{Y}|$ scaling is unnecessary for balancing the two terms. We therefore drop the constant scaling factors; this often leads to higher discrepancy. In practice we optimize this objective with multiple initializations and hyperparameters and select the solution with the largest empirical discrepancy on a holdout set to ensure a conservative bound. Experimentally, we find that replacing $\ell_{\mathrm{dis}}$ with any of the surrogate losses from [12, 50, 23] results in smaller discrepancy; we present these results in Appendix B.

**Tightening the bound by optimizing over the logits.** Looking at Theorem 3.6, it is clear that the value of the bound will decrease as the capacity of the hypothesis class is restricted. Since the number of features is large, one may expect that Assumption 3.5 holds even for a reduced feature set. In particular, it is well documented that deep networks optimized with stochastic gradient descent learn representations with small effective rank, often not much more than the number of classes [1, 2, 54, 29]. This suggests that the logits themselves should contain most of the features' information about $\mathcal{S}$ and $\mathcal{T}$ and that using the full feature space is unnecessarily conservative. To test this, we evaluate D$\mathrm{IS}^2$ on the full features, the logits output by $\hat{h}$, and various fractions of the top

principal components (PCs) of the features. We observe that using logits indeed results in tighter error bounds *while still remaining valid*—in contrast, using fewer top PCs also results in smaller error bounds, but at some point they become invalid (Figure C.2). The bounds we report in this work are thus evaluated on the logits of $\hat{h}$, except where we provide explicit comparisons in Section 5.

**Identifying the ideal number of PCs via a "validity score".** Even though reducing the feature dimensionality eventually results in an invalid bound, it is tempting to consider how we may identify approximately when this occurs, which could give a more accurate (though less conservative) prediction. We find that *the optimization trajectory itself* provides meaningful signal about this change. Specifically, Figure C.3 shows that for feature sets which are not overly restrictive, the critic very rapidly ascends to the maximum source agreement, then slowly begins overfitting. For much more restrictive feature sets (i.e., fewer PCs), the critic optimizes much more slowly, suggesting that we have reached the point where we are artificially restricting $\mathcal{H}$ and therefore underestimating the disagreement discrepancy. We design a "validity score" which captures this phenomenon, and we observe that it is roughly linearly correlated with the tightness of the eventual bound (Figure C.4). Though the score is by no means perfect, we can evaluate $\mathrm{DIS}^2$ with successively fewer PCs and only retain those above a certain score threshold, reducing the average prediction error while remaining reasonably conservative (Figure C.5). For further details, see Appendix C.

| Prediction Method | Coverage ($\uparrow$) | | Overest. ($\downarrow$) | | MAE ($\downarrow$) | |
| --- | --- | --- | --- | --- | --- | --- |
| **DA?** | ✗ | ✓ | ✗ | ✓ | ✗ | ✓ |
| AC [25] | $0.1000 \pm .032$ | $0.0333 \pm .023$ | $0.1194 \pm .012$ | $0.1123 \pm .012$ | $0.1091 \pm .011$ | $0.1091 \pm .012$ |
| DoC [24] | $0.1667 \pm .040$ | $0.0167 \pm .0167$ | $0.1237 \pm .012$ | $0.1096 \pm .012$ | $0.1055 \pm .011$ | $0.1083 \pm .012$ |
| ATC NE [21] | $0.2889 \pm .048$ | $0.1333 \pm .044$ | $0.0824 \pm .009$ | $0.0969 \pm .012$ | $0.0665 \pm .007$ | $0.0854 \pm .011$ |
| COT [40] | $0.2554 \pm .0467$ | $0.1667 \pm .049$ | $0.0860 \pm .009$ | $0.0948 \pm .011$ | $0.0700 \pm .007$ | $0.0808 \pm .010$ |
| $\mathrm{DIS}^2$ (Features) | $1.0000$ | $1.0000$ | $0.0000$ | $0.0000$ | $0.2807 \pm .009$ | $0.1918 \pm .008$ |
| $\mathrm{DIS}^2$ (Logits) | $0.9889 \pm .011$ | $0.7500 \pm .058$ | $0.0011 \pm .000$ | $0.0475 \pm .007$ | $0.1489 \pm .011$ | $0.0945 \pm .010$ |
| $\mathrm{DIS}^2$ (Logits w/o $\delta$) | $0.7556 \pm .0475$ | $0.4333 \pm .065$ | $0.0771 \pm .013$ | $0.0892 \pm .011$ | $0.0887 \pm .009$ | $0.0637 \pm .008$ |

Table 1: **Comparing the $\mathrm{DIS}^2$ bound to prior methods for predicting accuracy.** DA denotes if the representations were learned via a domain-adversarial algorithm. We report what fraction of predictions correctly bound the true error (Coverage) and the average prediction error among shifts whose accuracy is overestimated (Overest.), along with overall MAE. $\mathrm{DIS}^2$ has substantially higher coverage and lower overestimation error, though lower overall MAE. By dropping the concentration term in Theorem 3.6 we can get even better MAE—even beating the baselines on domain-adversarial representations—at some cost to coverage.

## 5 Experiments

**Datasets.** We conduct experiments across 11 vision benchmark datasets for distribution shift on datasets that span applications in object classification, satellite imagery, and medicine. We use four BREEDs datasets: [65] Entity13, Entity30, Nonliving26, and Living17; FMoW [11] and Camelyon [4] from WILDS [35]; Officehome [69]; Visda [52, 51]; CIFAR10, CIFAR100 [36]; and Domainet [53]. Each of these datasets consists of multiple domains with different types of natural and synthetic shifts. We consider subpopulation shift and natural shifts induced due to differences in the data collection process of ImageNet, i.e., ImageNetv2 [60] and a combination of both. For CIFAR10 and CIFAR100 we evaluate natural shifts due to variations in replication studies [59] and common corruptions [27]. For all datasets, we use the same source and target domains commonly used in previous studies [22, 64]. We provide precise details about the distribution shifts considered in Appendix A. Because distribution shifts vary widely in scope, prior evaluations which focus on only one specific type of shift (e.g., corruptions) often do not convey the full story. **We therefore emphasize the need for more comprehensive evaluations across many different types of shifts and training methods**, as we present here.

**Experimental setup and protocols.** Along with source-only training with ERM, we experiment with Unsupervised Domain Adaptation (UDA) methods that aim to improve target performance with unlabeled target data (FixMatch [66], DANN [18], CDAN [38], and BN-adapt [37]). We experiment with Densenet121 [28] and Resnet18/Resnet50 [26] pretrained on ImageNet. For source-only ERM, as with other methods, we default to using strong augmentations: random horizontal flips, random crops, as well as Cutout [16] and RandAugment [13]. Unless otherwise specified, we default to full finetuning for source-only ERM and UDA methods. We use source hold-out performance to pick the best hyperparameters for the UDA methods, since we lack labeled validation data from the target distribution. For all of these methods, we fix the algorithm-specific hyperparameters to their original recommendations following the experimental protocol in [22]. For more details, see Appendix A.

**Methods evaluated.** We compare $\text{DIS}^2$ to four competitive baselines: *Average Confidence* (AC; [25]), *Difference of Confidences* (DoC; [24]), *Average Thresholded Confidence* (ATC; [21]), and *Confidence Optimal Transport* (COT; [40]). We give detailed descriptions of these methods in Appendix A. For all methods, we implement post-hoc calibration on validation source data with temperature scaling [25], which has been shown to improve performance. For $\text{DIS}^2$, we report bounds evaluated both on the full features and on the logits of $\hat{h}$ as described in Section 4. Unless specified otherwise, we set $\delta = .01$ everywhere. We also experiment with dropping the lower order concentration term in Theorem 3.6, using only the sample average. Though this is of course no longer a conservative bound, we find it is an excellent predictor of test error and is worth including.

**Metrics for evaluation.** As our emphasis is on giving valid error bounds, we report the *coverage*, i.e. the fraction of predictions for which the true error does not exceed the predicted error. We also report the standard prediction metric, *mean absolute error* (MAE). Finally, we measure the *conditional average overestimation*: this is the MAE among predictions which overestimate the accuracy. This metric captures the idea that the most important thing is giving a valid bound—but if for some reason it is not, we'd at least like it to be as accurate as possible.

**Results.** Reported metrics for all methods can be found in Table 1. We aggregate results over all datasets, shifts, and training methods—we stratify only by whether the training method is domain-adversarial, as this affects the validity of Assumption 3.5. We find that $\text{DIS}^2$ achieves competitive MAE while maintaining substantially higher coverage, even for domain-adversarial features. When it does overestimate accuracy, it does so by much less, implying that it is ideal for conservative estimation even when any given error bound is not technically satisfied. Dropping the concentration term performs even better (sometimes beating the baselines), at the cost of some coverage. This suggests that efforts to better estimate the true maximum discrepancy may yield even better predictors. We also show scatter plots to visualize performance on individual distribution shifts, plotting each source-target pair as a single point. For these too we report separately the results for domain-adversarial (Figure 3) and non-domain-adversarial methods (Figure 1). To avoid clutter, these two plots do not include DoC, as it performed comparably to AC. Figure 4(a) displays additional scatter plots which allow for a direct comparison of the variants of $\text{DIS}^2$. Finally, Figure 4(b) plots the observed violation rate (i.e. $1 - $coverage) of $\text{DIS}^2$ on non-domain-adversarial methods for varying $\delta$. We observe that it lies at or below the line $y = x$, meaning the probabilistic bound provided by Theorem 3.6 holds across a range of failure probabilities. Thus we see that our probabilistic bound is empirically valid all of the time—not in the sense that each individual shift's error is upper bounded, but rather that the desired violation rate is always satisfied.

**Strengthening the baselines to improve coverage.** Since the baselines we consider in this work prioritize predictive accuracy over conservative estimates, their coverage can possibly be improved without too much increase in error. We explore this option using LOOCV: for a desired coverage, we learn a parameter to either scale or shift a method's prediction to achieve that level of coverage on all but one of the datasets. We then evaluate the method on all shifts of the remaining dataset, and we repeat this for each dataset. Appendix D reports the results for varying coverage levels. We find that (i) the baselines do not achieve the desired coverage on the held out data, though they get somewhat close; and (ii) the adjustment causes them to suffer higher MAE than $\text{DIS}^2$. Thus $\text{DIS}^2$ is on the Pareto frontier of MAE and coverage, and is preferable when conservative bounds are desirable. We believe identifying alternative methods of post-hoc prediction adjustment is a promising future direction.

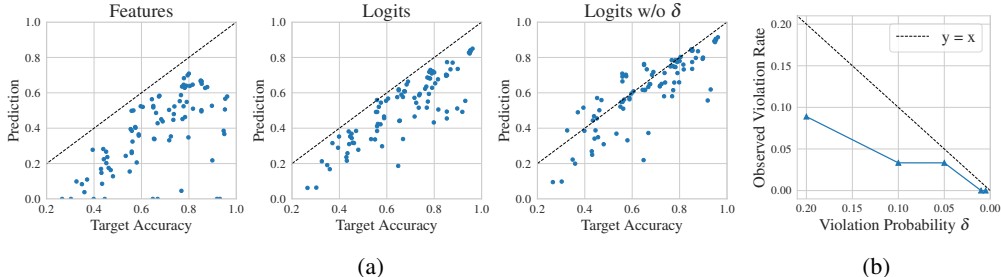

Figure 4: **(a):** Scatter plots depicting $\text{DIS}^2$ estimated bound vs. true error for a variety of shifts. "w/o $\delta$" indicates that the lower-order term of Theorem 3.6 has been dropped. **(b):** Observed bound violation rate vs. desired probability $\delta$. Observe that the true rate lies at or below $y = x$ across a range of values.

## 6 Conclusion

The ability to evaluate *trustworthy*, non-vacuous error bounds for deep neural networks under distribution shift remains an extremely important open problem. Due to the wide variety of real-world shifts and the complexity of modern data, restrictive a priori assumptions on the distribution (i.e., before observing any data from the shift of interest) seem unlikely to be fruitful. On the other hand, prior methods which estimate accuracy using extra information—such as unlabeled test samples— often rely on opaque conditions whose likelihood of being satisfied is difficult to predict, and so they sometimes provide large underestimates of test error with no warning signs.

This work bridges this gap with a simple, intuitive condition and a new disagreement loss which together result in competitive error *prediction*, while simultaneously providing an (almost) guaranteed probabilistic error *bound*. We also study how the process of evaluating the bound (e.g., the optimization landscape) can provide even more useful signal, enabling better predictive accuracy. We expect there is potential to push further in each of these directions, hopefully extending the current accuracy-reliability Pareto frontier for test error bounds under distribution shift.

## Acknowledgments and Disclosure of Funding

Thanks to Sam Sokota, Bingbin Liu, Yuchen Li, Yiding Jiang, Zack Lipton, Roni Rosenfeld, and Andrej Risteski for helpful comments. ER acknowledges the support of NSF via IIS-1909816, IIS-1955532, OAC-1934584. SG acknowledges Amazon Graduate Fellowship and JP Morgan AI Ph.D. Fellowship for their support.

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

# Appendix

## A  Experimental Details

### A.1  Description of Baselines

*Average Thresholded Confidence (ATC).*  ATC first estimates a threshold $t$ on the confidence of softmax prediction (or on negative entropy) such that the number of source labeled points that get a confidence greater than $t$ match the fraction of correct examples, and then estimates the test error on on the target domain $\mathcal{D}_{\text{test}}$ as the expected number of target points that obtain a score less than $t$, i.e.,

$$\text{ATC}_{\mathcal{D}_{\text{test}}}(s) = \sum_{i=1}^{n} \mathbb{I}\left[s(f(x_i')) < t\right],$$

where $t$ satisfies: $\sum_{i=1}^{j} \mathbb{I}\left[\max_{j \in \mathcal{Y}}(f_j(x_i)) < t\right] = \sum_{i=1}^{m} \mathbb{I}\left[\arg\max_{j \in \mathcal{Y}} f_j(x_i) \neq y_i\right]$

*Average Confidence (AC).*  Error is estimated as the average value of the maximum softmax confidence on the target data, i.e, $\text{AC}_{\mathcal{D}_{\text{test}}} = \sum_{i=1}^{n} \max_{j \in \mathcal{Y}} f_j(x_i')$.

*Difference Of Confidence (DOC).*  We estimate error on the target by subtracting the difference of confidences on source and target (as a surrogate to distributional distance [24]) from the error on source distribution, i.e., $\text{DOC}_{\mathcal{D}_{\text{test}}} = \sum_{i=1}^{n} \max_{j \in \mathcal{Y}} f_j(x_i') + \sum_{i=1}^{m} \mathbb{I}\left[\arg\max_{j \in \mathcal{Y}} f_j(x_i) \neq y_i\right] - \sum_{i=1}^{m} \max_{j \in \mathcal{Y}} f_j(x_i)$. This is referred to as DOC-Feat in [24].

*Confidence Optimal Transport (COT).*  COT uses the empirical estimator of the Earth Mover's Distance between labels from the source domain and softmax outputs of samples from the target domain to provide accuracy estimates:

$$\text{COT}_{\mathcal{D}_{\text{test}}}(s) = \frac{1}{2} \min_{\pi \in \Pi(S^n, Y^m)} \sum_{i,j=1}^{n,m} \left\| s_i - e_{y_j} \right\|_2 \pi_{ij},$$

where $S^n = \{f(x_i')\}_{i=1}^{n}$ are the softmax outputs on the unlabeled target data and $Y^m = \{y_j\}_{j=1}^{m}$ are the labels on holdout source examples.

For all of the methods described above, we assume that $\{(x_i')\}_{i=1}^{n}$ are the unlabeled target samples and $\{(x_i, y_i)\}_{i=1}^{m}$ are hold-out labeled source samples.

### A.2  Dataset Details

In this section, we provide additional details about the datasets used in our benchmark study.

- **CIFAR10**  We use the original CIFAR10 dataset [36] as the source dataset. For target domains, we consider (i) synthetic shifts (CIFAR10-C) due to common corruptions [27]; and (ii) natural distribution shift, i.e., CIFAR10v2 [58, 68] due to differences in data collection strategy. We randomly sample 3 set of CIFAR-10-C datasets. Overall, we obtain 5 datasets (i.e., CIFAR10v1, CIFAR10v2, CIFAR10C-Frost (severity 4), CIFAR10C-Pixelate (severity 5), CIFAR10-C Saturate (severity 5)).
- **CIFAR100**  Similar to CIFAR10, we use the original CIFAR100 set as the source dataset. For target domains we consider synthetic shifts (CIFAR100-C) due to common corruptions. We sample 4 CIFAR100-C datasets, overall obtaining 5 domains (i.e., CIFAR100, CIFAR100C-Fog (severity 4), CIFAR100C-Motion Blur (severity 2), CIFAR100C-Contrast (severity 4), CIFAR100C-spatter (severity 2) ).
- **FMoW**  In order to consider distribution shifts faced in the wild, we consider FMoW-WILDs [35, 11] from WILDS benchmark, which contains satellite images taken in different geographical regions and at different times. We use the original train as source and OOD val and OOD test splits as target domains as they are collected over different time-period. Overall, we obtain 3 different domains.
- **Camelyon17**  Similar to FMoW, we consider tumor identification dataset from the wilds benchmark [4]. We use the default train as source and OOD val and OOD test splits as target domains as they are collected across different hospitals. Overall, we obtain 3 different domains.

- **BREEDs**  We also consider BREEDs benchmark [65] in our setup to assess robustness to subpopulation shifts. BREEDs leverage class hierarchy in ImageNet to re-purpose original classes to be the subpopulations and defines a classification task on superclasses. We consider distribution shift due to subpopulation shift which is induced by directly making the subpopulations present in the training and test distributions disjoint. BREEDs benchmark contains 4 datasets **Entity-13**, **Entity-30**, **Living-17**, and **Non-living-26**, each focusing on different subtrees and levels in the hierarchy. We also consider natural shifts due to differences in the data collection process of ImageNet [63], e.g, ImageNetv2 [60] and a combination of both. Overall, for each of the 4 BREEDs datasets (i.e., Entity-13, Entity-30, Living-17, and Non-living-26), we obtain four different domains. We refer to them as follows: BREEDsv1 sub-population 1 (sampled from ImageNetv1), BREEDsv1 sub-population 2 (sampled from ImageNetv1), BREEDsv2 sub-population 1 (sampled from ImageNetv2), BREEDsv2 sub-population 2 (sampled from ImageNetv2). For each BREEDs dataset, we use BREEDsv1 sub-population A as source and the other three as target domains.
- **OfficeHome**  We use four domains (art, clipart, product and real) from OfficeHome dataset [69]. We use the product domain as source and the other domains as target.
- **DomainNet**  We use four domains (clipart, painting, real, sketch) from the Domainnet dataset [53]. We use real domain as the source and the other domains as target.
- **Visda**  We use three domains (train, val and test) from the Visda dataset [52]. While 'train' domain contains synthetic renditions of the objects, 'val' and 'test' domains contain real world images. To avoid confusing, the domain names with their roles as splits, we rename them as 'synthetic', 'Real-1' and 'Real-2'. We use the synthetic (original train set) as the source domain and use the other domains as target.

### A.3  Setup and Protocols

**Architecture Details**  For all datasets, we used the same architecture across different algorithms:

- CIFAR-10: Resnet-18 [26] pretrained on Imagenet
- CIFAR-100: Resnet-18 [26] pretrained on Imagenet
- Camelyon: Densenet-121 [28] *not* pretrained on Imagenet as per the suggestion made in [35]
- FMoW: Densenet-121 [28] pretrained on Imagenet
- BREEDs (Entity13, Entity30, Living17, Nonliving26): Resnet-18 [26] *not* pretrained on Imagenet as per the suggestion in [65]. The main rationale is to avoid pre-training on the superset dataset where we are simulating sub-population shift.
- Officehome: Resnet-50 [26] pretrained on Imagenet
- Domainnet: Resnet-50 [26] pretrained on Imagenet
- Visda: Resnet-50 [26] pretrained on Imagenet

Except for Resnets on CIFAR datasets, we used the standard pytorch implementation [19]. For Resnet on cifar, we refer to the implementation here: `https://github.com/kuangliu/pytorch-cifar`. For all the architectures, whenever applicable, we add antialiasing [71]. We use the official library released with the paper.

For imagenet-pretrained models with standard architectures, we use the publicly available models here: `https://pytorch.org/vision/stable/models.html`. For imagenet-pretrained models on the reduced input size images (e.g. CIFAR-10), we train a model on Imagenet on reduced input size from scratch. We include the model with our publicly available repository.

**Hyperparameter details**  First, we tune learning rate and $\ell_2$ regularization parameter by fixing batch size for each dataset that correspond to maximum we can fit to 15GB GPU memory. We set the number of epochs for training as per the suggestions of the authors of respective benchmarks. Note that we define the number of epochs as a full pass over the labeled training source data. We summarize learning rate, batch size, number of epochs, and $\ell_2$ regularization parameter used in our study in Table A.3.

For each algorithm, we use the hyperparameters reported in the initial papers. For domain-adversarial methods (DANN and CDANN), we refer to the suggestions made in Transfer Learning Library [31]. We tabulate hyperparameters for each algorithm next:

| Dataset | Source | Target |
|---|---|---|
| CIFAR10 | CIFAR10v1 | CIFAR10v1, CIFAR10v2, CIFAR10C-Frost (severity 4), CIFAR10C-Pixelate (severity 5), CIFAR10-C Saturate (severity 5) |
| CIFAR100 | CIFAR100 | CIFAR100, CIFAR100C-Fog (severity 4), CIFAR100C-Motion Blur (severity 2), CIFAR100C-Contrast (severity 4), CIFAR100C-spatter (severity 2) |
| Camelyon | Camelyon (Hospital 1–3) | Camelyon (Hospital 1–3), Camelyon (Hospital 4), Camelyon (Hospital 5) |
| FMoW | FMoW (2002–'13) | FMoW (2002–'13), FMoW (2013–'16), FMoW (2016–'18) |
| Entity13 | Entity13 (ImageNetv1 sub-population 1) | Entity13 (ImageNetv1 sub-population 1), Entity13 (ImageNetv1 sub-population 2), Entity13 (ImageNetv2 sub-population 1), Entity13 (ImageNetv2 sub-population 2) |
| Entity30 | Entity30 (ImageNetv1 sub-population 1) | Entity30 (ImageNetv1 sub-population 1), Entity30 (ImageNetv1 sub-population 2), Entity30 (ImageNetv2 sub-population 1), Entity30 (ImageNetv2 sub-population 2) |
| Living17 | Living17 (ImageNetv1 sub-population 1) | Living17 (ImageNetv1 sub-population 1), Living17 (ImageNetv1 sub-population 2), Living17 (ImageNetv2 sub-population 1), Living17 (ImageNetv2 sub-population 2) |
| Nonliving26 | Nonliving26 (ImageNetv1 sub-population 1) | Nonliving26 (ImageNetv1 sub-population 1), Nonliving26 (ImageNetv1 sub-population 2), Nonliving26 (ImageNetv2 sub-population 1), Nonliving26 (ImageNetv2 sub-population 2) |
| Officehome | Product | Product, Art, ClipArt, Real |
| DomainNet | Real | Real, Painiting, Sketch, ClipArt |
| Visda | Synthetic (originally referred to as train) | Synthetic, Real-1 (originally referred to as val), Real-2 (originally referred to as test) |

Table A.2: Details of the source and target datasets in our testbed.

| Dataset | Epoch | Batch size | $\ell_2$ regularization | Learning rate |
|---|---|---|---|---|
| CIFAR10 | 50 | 200 | 0.0001 (chosen from $\{0.0001, 0.001, 1e\text{-}5, 0.0\}$) | 0.01 (chosen from $\{0.001, 0.01, 0.0001\}$) |
| CIFAR100 | 50 | 200 | 0.0001 (chosen from $\{0.0001, 0.001, 1e\text{-}5, 0.0\}$) | 0.01 (chosen from $\{0.001, 0.01, 0.0001\}$) |
| Camelyon | 10 | 96 | 0.01 (chosen from $\{0.01, 0.001, 0.0001, 0.0\}$) | 0.03 (chosen from $\{0.003, 0.3, 0.0003, 0.03\}$) |
| FMoW | 30 | 64 | 0.0 (chosen from $\{0.0001, 0.001, 1e\text{-}5, 0.0\}$) | 0.0001 (chosen from $\{0.001, 0.01, 0.0001\}$) |
| Entity13 | 40 | 256 | 5e-5 (chosen from $\{5e\text{-}5, 5e\text{-}4, 1e\text{-}4, 1e\text{-}5\}$) | 0.2 (chosen from $\{0.1, 0.5, 0.2, 0.01, 0.0\}$) |
| Entity30 | 40 | 256 | 5e-5 (chosen from $\{5e\text{-}5, 5e\text{-}4, 1e\text{-}4, 1e\text{-}5\}$) | 0.2 (chosen from $\{0.1, 0.5, 0.2, 0.01, 0.0\}$) |
| Living17 | 40 | 256 | 5e-5 (chosen from $\{5e\text{-}5, 5e\text{-}4, 1e\text{-}4, 1e\text{-}5\}$) | 0.2 (chosen from $\{0.1, 0.5, 0.2, 0.01, 0.0\}$) |
| Nonliving26 | 40 | 256 | 0 5e-5 (chosen from $\{5e\text{-}5, 5e\text{-}4, 1e\text{-}4, 1e\text{-}5\}$) | 0.2 (chosen from $\{0.1, 0.5, 0.2, 0.01, 0.0\}$) |
| Officehome | 50 | 96 | 0.0001 (chosen from $\{0.0001, 0.001, 1e\text{-}5, 0.0\}$) | 0.01 (chosen from $\{0.001, 0.01, 0.0001\}$) |
| DomainNet | 15 | 96 | 0.0001 (chosen from $\{0.0001, 0.001, 1e\text{-}5, 0.0\}$) | 0.01 (chosen from $\{0.001, 0.01, 0.0001\}$) |
| Visda | 10 | 96 | 0.0001 (chosen from $\{0.0001, 0.001, 1e\text{-}5, 0.0\}$) | 0.01 (chosen from $\{0.001, 0.01, 0.0001\}$) |

Table A.3: Details of the learning rate and batch size considered in our testbed

- **DANN, CDANN,** As per Transfer Learning Library suggestion, we use a learning rate multiplier of 0.1 for the featurizer when initializing with a pre-trained network and 1.0 otherwise. We default to a penalty weight of 1.0 for all datasets with pre-trained initialization.

- **FixMatch**  We use the lambda is 1.0 and use threshold $\tau$ as 0.9.

**Compute Infrastructure**  Our experiments were performed across a combination of Nvidia T4, A6000, and V100 GPUs.

## B  Comparing Disagreement Losses

We define the alternate losses for maximizing disagreement:

1. Chuang et al. [12] minimize the negative cross-entropy loss, which is concave in the model logits. That is, they add the term $\log \operatorname{softmax}(h(x)_y)$ to the objective they are minimizing. This loss results in substantially lower disagreement discrepancy than the other two.
2. Pagliardini et al. [50] use a loss which is not too different from ours. They define the disagreement objective for a point $(x, y)$ as

$$\log\left(1 + \frac{\exp(h(x)_y)}{\sum_{\hat{y} \neq y} \exp(h(x)_{\hat{y}})}\right). \tag{1}$$

For comparison, $\ell_{\text{dis}}$ can be rewritten as

$$\log\left(1 + \frac{\exp(h(x)_y)}{\exp\left(\frac{1}{|\mathcal{Y}|-1} \sum_{\hat{y} \neq y} h(x)_{\hat{y}}\right)}\right), \tag{2}$$

where the incorrect logits are averaged and the exponential is pushed outside the sum. This modification results in (2) being convex in the logits and an upper bound to the disagreement 0-1 loss, whereas (1) is neither.

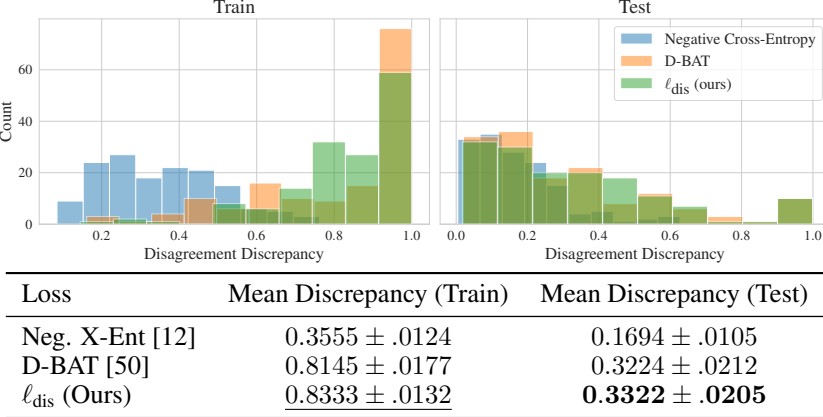

| Loss | Mean Discrepancy (Train) | Mean Discrepancy (Test) |
| --- | --- | --- |
| Neg. X-Ent [12] | $0.3555 \pm .0124$ | $0.1694 \pm .0105$ |
| D-BAT [50] | $0.8145 \pm .0177$ | $0.3224 \pm .0212$ |
| $\ell_{\text{dis}}$ (Ours) | $\underline{0.8333 \pm .0132}$ | $\mathbf{0.3322 \pm .0205}$ |

Figure B.1 & Table B.3: Histogram of disagreement discrepancies for each of the three losses, and the average values across all datasets. **Bold** (resp. Underline) indicates the method has higher average discrepancy under a paired t-test at significance $p = .01$ (resp. $p = .025$).

Figure B.1 displays histograms of the achieved disagreement discrepancy across all distributions for each of the disagreement losses (all hyperparameters and random seeds are the same for all three losses). The table below it reports the mean disagreement discrepancy on the train and test sets. We find that the negative cross-entropy, being a concave function, results in very low discrepancy. The D-BAT loss (Equation (1)) is reasonably competitive with our loss (Equation (2)) on average, seemingly because it gets very high discrepancy on a subset of shifts. This suggests that it may be particularly suited for a specific type of distribution shift, though it is less good overall. Though the averages are reasonably close, the samples are not independent, so we run a paired t-test and we find that the increases to average train and test discrepancies achieved by $\ell_{\text{dis}}$ are significant at levels $p = 0.024$ and $p = 0.009$, respectively. With enough holdout data, a reasonable approach would be to split the data in two: one subset to validate critics trained on either of the two losses, and another to evaluate the discrepancy of whichever one is ultimately selected.

# C    Exploration of the Validity Score

To experiment with reducing the complexity of the class $\mathcal{H}$, we evaluate $\text{DIS}^2$ on progressively fewer top principal components (PCs) of the features. Precisely, for features of dimension $d$, we evaluate $\text{DIS}^2$ on the same features projected onto their top $d/k$ components, for $k \in [1, 4, 16, 32, 64, 128]$ (Figure C.2). We see that while projecting to fewer and fewer PCs does reduce the error bound value, unlike the logits it is a rather crude way to reduce complexity of $\mathcal{H}$, meaning at some point it goes too far and results in invalid error bounds.

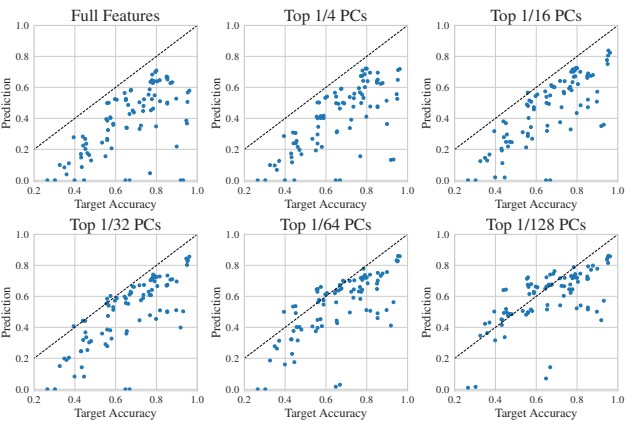

Figure C.2: **$\text{DIS}^2$ bound as fewer principal components are kept.** Reducing the number of top principal components crudely reduces complexity of $\mathcal{H}$—this leads to lower error estimates, but at some point the bounds become invalid for a large fraction of shifts.

However, during the optimization process we observe that around when this violation occurs, the task of training a critic to both agree on $\mathcal{S}$ and disagree on $\mathcal{T}$ goes from "easy" to "hard". Figure C.3 shows that on the full features, the critic rapidly ascends to maximum agreement on $\mathcal{S}$, followed by slow decay (due to both overfitting and learning to simultaneously disagree on $\mathcal{T}$). As we drop more and more components, this optimization becomes slower.

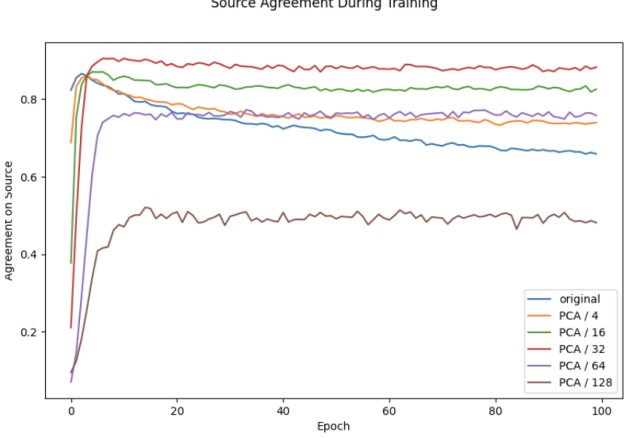

Figure C.3: **Agreement on one shift between $\hat{h}$ and $h'$ on $\hat{\mathcal{S}}$ during optimization.** We observe that as the number of top PCs retained drops, the optimization occurs more slowly and less monotonically. For this particular shift, the bound becomes invalid when keeping only the top $1/128$ components, depicted by the brown line.

We therefore design a "validity score" intended to capture this phenomenon which we refer to as the *cumulative $\ell_1$ ratio*. This is defined as the maximum agreement achieved, divided by the cumulative sum of absolute differences in agreement across all epochs up until the maximum was achieved.

Formally, let $\{a_i\}_{i=1}^T$ represent the agreement between $h'$ and $\hat{h}$ after epoch $i$, i.e. $1 - \epsilon_{\hat{S}}(\hat{h}, h_i')$, and define $m := \arg\max_{i \in [T]} a_i$. The cumulative $\ell_1$ ratio is then $\frac{a_m}{a_1 + \sum_{i=2}^m |a_i - a_{i-1}|}$. Thus, if the agreement rapidly ascends to its maximum without ever going down over the course of an epoch, this ratio will be equal to 1, and if it non-monotonically ascends then the ratio will be significantly less. This definition was simply the first metric we considered which approximately captures the behavior we observed; we expect it could be greatly improved.

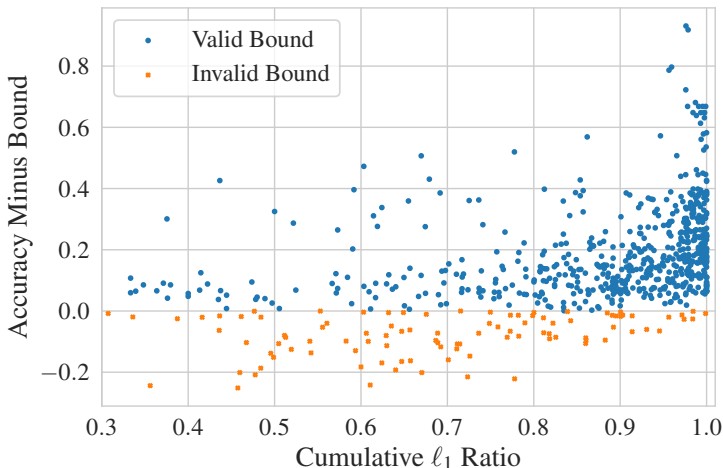

Figure C.4: **Cumulative $\ell_1$ ratio versus error prediction gap.** Despite its simplicity, the ratio captures the information encoded in the optimization trajectory, roughly linearly correlating with the tightness and validity of a given prediction. It is thus a useful metric for identifying the ideal number of top PCs to use.

Figure C.4 displays a scatter plot of the cumulative $\ell_1$ ratio versus the difference in estimated and true error for $\mathrm{D_{IS}}^2$ evaluated on the full range of top PCs. A negative value implies that we have underestimated the error (i.e., the bound is not valid). We see that even this very simply metric roughly linearly correlates with the tightness of the bound, which suggests that evaluating over a range of top PC counts and only keeping predictions whose $\ell_1$ ratio is above a certain threshold can improve raw predictive accuracy without reducing coverage by too much. Figure C.5 shows that this is indeed the case: compared to $\mathrm{D_{IS}}^2$ evaluated on the logits, keeping all predictions above a score threshold can produce more accurate error estimates, without *too* severely underestimating error in the worst case.

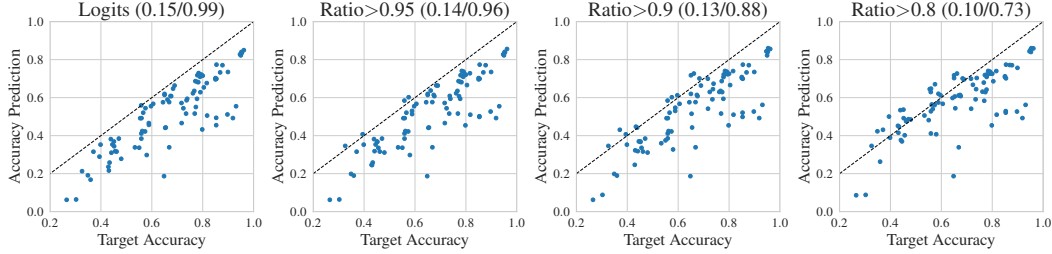

Figure C.5: $\mathrm{D_{IS}}^2$ **bounds and MAE / coverage as the cumulative $\ell_1$ ratio threshold is lowered.** Values in parenthesis are (MAE / coverage). By only keeping predictions with ratio above a varying threshold, we can smoothly interpolate between bound validity and raw error prediction accuracy.

# D    Making Baselines More Conservative with LOOCV

To more thoroughly compare $\mathrm{DIS}^2$ to prior estimation techniques, we consider a strengthening of the baselines which may give them higher coverage without too much cost to prediction accuracy. Specifically, for each desired coverage level $\alpha \in [0.9, 0.95, 0.99]$, we use all but one of the datasets to learn a parameter to either scale or shift a method's predictions enough to achieve coverage $\alpha$. We then evaluate this scaled or shifted prediction on the distribution shifts of the remaining dataset, and we repeat this for each one.

The results, found in Table D.4, demonstrate that prior methods can indeed be made to have much higher coverage, although as expected their MAE suffers. Furthermore, they still underestimate error on the tail distribution shifts by quite a bit, and they rarely achieve the desired coverage on the heldout dataset—though they usually come reasonably close. In particular, ATC [21] and COT [40] do well with a shift parameter, e.g. at the desired coverage $\alpha = 0.95$ ATC matches $\mathrm{DIS}^2$ in MAE and gets 94.4% coverage (compared to 98.9% by $\mathrm{DIS}^2$). However, its conditional average overestimation is quite high, almost 9%. COT gets much lower overestimation (particularly for higher coverage levels), and it also appears to suffer less on the tail distribution shifts in the sense that $\alpha = 0.99$ does not induce nearly as high MAE as it does for ATC. However, at that level it only achieves 95.6% coverage, and it averages almost 5% accuracy overestimation on the shifts it does not correctly bound (compared to 0.1% by $\mathrm{DIS}^2$). Also, its MAE is still substantially higher than $\mathrm{DIS}^2$, despite getting lower coverage. Finally, we evaluate the scale/shift approach on our $\mathrm{DIS}^2$ bound without the lower order term, but based on the metrics we report there appears to be little reason to prefer it over the untransformed version, one of the baselines, or the original $\mathrm{DIS}^2$ bound.

Taken together, these results imply that if one's goal is predictive accuracy and tail behavior is not important (worst ~10%), ATC or COT will likely get reasonable coverage with a shift parameter—though they still significantly underestimate error on a non-negligible fraction of shifts. If one cares about the long tail of distribution shifts, or prioritizes being conservative at a slight cost to average accuracy, $\mathrm{DIS}^2$ is clearly preferable. Finally, we observe that the randomness which determines which shifts are not correctly bounded by $\mathrm{DIS}^2$ is "decoupled" from the distributions themselves under Theorem 3.6, in the sense that it is an artifact of the random samples, rather than a property of the distribution (recall Figure 4(b)). This is in contrast with the shift/scale approach which would produce almost identical results under larger sample sizes because it does not account for finite sample effects. This implies that some distribution shifts are simply "unsuitable" for prior methods because they do not satisfy whatever condition these methods rely on, and observing more samples will not remedy this problem. It is clear that working to understand these conditions is crucial for reliability and interpretability, since we are not currently able to identify which distributions are suitable a priori.

| Method | Adjustment | MAE (↓) | | | Coverage (↑) | | | Overest. (↓) | | |
| --- | --- | --- | --- | --- | --- | --- | --- | --- | --- | --- |
| $\alpha \rightarrow$ | | 0.9 | 0.95 | 0.99 | 0.9 | 0.95 | 0.99 | 0.9 | 0.95 | 0.99 |
| AC | none | | 0.106 | | | 0.122 | | | 0.118 | |
| | shift | 0.153 | 0.201 | 0.465 | 0.878 | 0.922 | 0.956 | 0.119 | 0.138 | 0.149 |
| | scale | 0.195 | 0.221 | 0.416 | 0.911 | 0.922 | 0.967 | 0.135 | 0.097 | 0.145 |
| DoC | none | | 0.105 | | | 0.167 | | | 0.122 | |
| | shift | 0.158 | 0.200 | 0.467 | 0.878 | 0.911 | 0.956 | 0.116 | 0.125 | 0.154 |
| | scale | 0.195 | 0.223 | 0.417 | 0.900 | 0.944 | 0.967 | 0.123 | 0.139 | 0.139 |
| ATC NE | none | | 0.067 | | | 0.289 | | | 0.083 | |
| | shift | 0.117 | 0.150 | 0.309 | 0.900 | 0.944 | 0.978 | 0.072 | 0.088 | 0.127 |
| | scale | 0.128 | 0.153 | 0.357 | 0.889 | 0.933 | 0.978 | 0.062 | 0.074 | 0.144 |
| COT | none | | 0.069 | | | 0.256 | | | 0.085 | |
| | shift | 0.115 | 0.140 | 0.232 | 0.878 | 0.944 | 0.956 | 0.049 | 0.065 | 0.048 |
| | scale | 0.150 | 0.193 | 0.248 | 0.889 | 0.944 | 0.956 | 0.074 | 0.066 | 0.044 |
| $\mathrm{D_{IS}}^2$ (w/o $\delta$) | none | | 0.083 | | | 0.756 | | | 0.072 | |
| | shift | 0.159 | 0.169 | 0.197 | 0.889 | 0.933 | 0.989 | 0.021 | 0.010 | 0.017 |
| | scale | 0.149 | 0.168 | 0.197 | 0.889 | 0.933 | 0.989 | 0.023 | 0.021 | 0.004 |
| $\mathrm{D_{IS}}^2$ ($\delta = 10^{-2}$) | none | | 0.150 | | | 0.989 | | | 0.001 | |
| $\mathrm{D_{IS}}^2$ ($\delta = 10^{-3}$) | none | | 0.174 | | | 1.000 | | | 0.000 | |

Table D.4: MAE, coverage, and conditional average overestimation for the strengthened baselines with a shift or scale parameter on non-domain-adversarial representations. Because a desired coverage $\alpha$ is only used when an adjustment is learned, "none"—representing no adjustment—does not vary with $\alpha$.

# E   Proving that Assumption 3.5 Holds for Some Datasets

Here we describe how the equivalence of Assumption 3.5 and the bound in Theorem 3.6 allow us to prove that the assumption holds with high probability. By repeating essentially the same proof as Theorem 3.6 in the other direction, we get the following corollary:

**Corollary E.1.** *If Assumption 3.5 does* not *hold, then with probability $\geq 1 - \delta$,*

$$\epsilon_{\hat{\mathcal{T}}}(\hat{h}) > \epsilon_{\hat{\mathcal{S}}}(\hat{h}) + \hat{\Delta}(\hat{h}, h') - \sqrt{\frac{2(n_S + n_T)\log{1/\delta}}{n_S n_T}}.$$

Note that the concentration term here is different from Theorem 3.6 because we are bounding the empirical target error, rather than the true target error. The reason for this change is that now we can make direct use of its contrapositive:

**Corollary E.2.** *With probability $\geq 1 - \delta$ over the randomness of the samples $\hat{\mathcal{S}}$ and $\hat{\mathcal{T}}$, if it is the case that*

$$\epsilon_{\hat{\mathcal{T}}}(\hat{h}) \leq \epsilon_{\hat{\mathcal{S}}}(\hat{h}) + \hat{\Delta}(\hat{h}, h') - \sqrt{\frac{2(n_S + n_T)\log{1/\delta}}{n_S n_T}},$$

*then Assumption 3.5 must hold.*

We evaluate this bound on non-domain-adversarial shifts with $\delta = 10^{-6}$. As some of the BREEDS shifts have as few as 68 test samples, we restrict ourselves to shifts with $n_T \geq 500$ to ignore those where the finite-sample term heavily dominates; this removes a little over 20% of all shifts. Among the remainder, we find that the bound in Corollary E.2 holds 55.7% of the time when using full features and 25.7% of the time when using logits. This means that for these shifts, we can be essentially certain that Assumption 3.5—and therefore also Assumption 3.3—is true.

Note that the fact that the bound is *not* violated for a given shift does not at all imply that the assumption is not true. In general, the only rigorous way to prove that Assumption 3.5 does not hold would be to show that for a fixed $\delta$, the fraction of shifts for which the bound in Theorem 3.6 does not hold is larger than $\delta$ (in a manner that is statistically significant under the appropriate hypothesis test). Because this never occurs in our experiments, we cannot conclude that the assumption is ever false. At the same time, the fact that the bound *does* hold at least $1 - \delta$ of the time does not prove that the assumption is true—it merely suggests that it is reasonable and that the bound should continue to hold in the future. This is why it is important for Assumption 3.5 to be simple and intuitive, so that we can trust that it will persist and anticipate when it will not.

However, Corollary E.2 allows us to make a substantially stronger statement. In fact, it says that for *any* distribution shift, with enough samples, we can prove a posteriori whether or not Assumption 3.5 holds, because the gap between these two bounds will shrink with increasing sample size.

## F    Figure 1 Stratified by Training Method

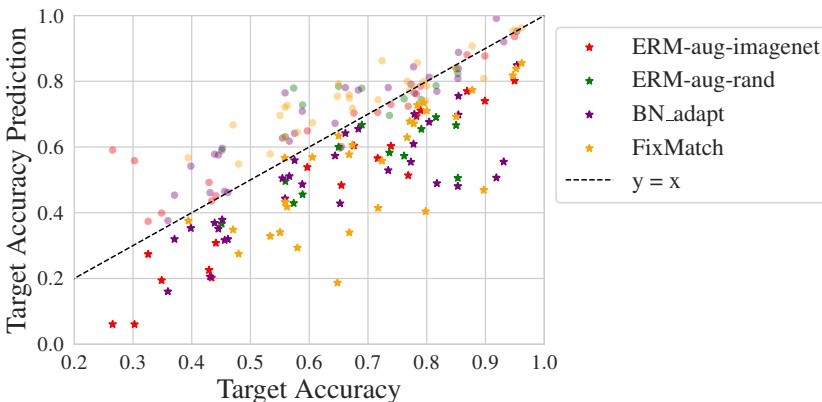

Figure F.6: **Error prediction stratified by training method.** Stars denote $\textsc{Dis}^2$, circles are ATC NE. We see that $\textsc{Dis}^2$ maintains its validity across different training methods.

