# OpenReview forum: "(Almost) Provable Error Bounds Under Distribution Shift via Disagreement Discrepancy"
_NeurIPS.cc/2023/Conference — NeurIPS 2023 poster_

### Official Review · Reviewer_PQBq · 2023-07-04

**Soundness:** 1 poor
**Presentation:** 2 fair
**Contribution:** 2 fair
**Rating:** 3
**Confidence:** 3

**Summary:**

This paper try to propose a new metric on evaluation of the distribution shift of dataset. The authors first prove the proposed error estimation bound with some assumptions. Then they demonstrate the effectiveness of their method through training a surrogate model maximizing the disagreement discrepancy. The authors conduct experiments on different dataset, showing that their method never overestimate the prediction accuracy on the target domain.

**Strengths:**

1. The authors try to use unlabeled data in the target domain, which is believed can bring benefits on predict the accuracy on the target domain.
2. The author introduce an interesting conception, named disagreement discrepancy, that represents the maximum difference between the target domain and source domain in the hypothesis set. This conception may provide some insights on the investigation of domain shift.



**Weaknesses:**

1. Certain evaluation metrics seem unclear. The reviewer finds it confusing why the method's characteristic of never overestimating the target accuracy—consistently underestimating it—is considered advantageous, particularly when its Mean Absolute Error (MAE) significantly underperforms compared to other methods. The reviewer recommends the authors to illustrate the benefits of this particular feature.
2. When compared with the baseline methods, the proposed metric falls short in terms of some crucial metrics, such as the Mean Absolute Error (MAE)
3. The authors attempt to train an additional network by maximizing the discrepancy in disagreement. However, this approach may be time-consuming and unreliable due to its sensitivity to the training hyperparameters.

**Questions:**

Most questions are listed in the weakness part.

Minor:
1. Please explain the meaning of '(almost) Provable'

**Limitations:**

Shown in weakness part

---

> ### Author Rebuttal · Authors · 2023-08-10
>
> We thank the reviewer for their feedback.
>
> In your review, you evaluated the soundness as “poor”. We are not sure we understand the reason for this evaluation. Our understanding is that soundness is meant to evaluate the correctness of claims in a given work. Do you believe that our bound or our experiments are not valid? **If not, could you please clarify why you gave this evaluation, so that we can address whatever concerns you have?**
>
>
> * “**The reviewer finds it confusing why…never overestimating the target accuracy…is considered advantageous, particularly when MAE significantly underperforms.**
>
> Consider a scenario where a model will be making crucial decisions: self-driving cars, medical diagnoses, financial agents, etc. If we deploy this model expecting 80% accuracy and its actual accuracy is 20%, the result could be very costly. In this setting, a valid error bound is *essential* and accuracy of an error prediction only matters if we can trust it.
>
> This is exactly when consistent error overestimation is important. Say we need a *maximum* of 20% error for a model to be reasonably safely deployed. If our method predicts 20% error, you can be quite sure that it’s ok to deploy. If our method predicts 50% error, the model *might* still be ok to deploy—but we probably don’t want to take that chance!
>
> If you were to use a different method to predict error here and it predicted 10% error, you might then deploy it, only to discover that the *true* error is 50%. **But by then the damage may already be done.**
>
> We hope it is clear that there are *many* settings where overestimating error is advantageous. Please see our response to all reviewers for more details.
>
> > **”compared with the baseline methods, the proposed metric falls short in terms of some crucial metrics, such as the Mean Absolute Error (MAE).**
>
> We want to emphasize that **the primary focus of this work is on giving valid, non-vacuous error *bounds***, with accurate prediction being secondary, though still important. We believe we were as upfront as possible about this in our writeup. Indeed, **as early as the abstract we state that we do not beat the baselines purely on MAE.**
>
> Where our method *does* outperform existing methods is reliability, and it does so substantially. Based on our point above about the need for reliability, we hope you agree that this also represents a meaningful contribution. We discuss this in greater detail in our response to all reviewers.
>
> You stated that our method falls short on metric***s*** (plural). But we are not sure what other metrics you may be referring to? Our method does *much* better on coverage and has much lower MAE on the (very few) points for which it underestimates error. **Could you please clarify what other metrics you were talking about?**
>
>
> > **”The authors…train an additional network by maximizing the discrepancy in disagreement. However, this approach may be time-consuming and unreliable due to its sensitivity to the training hyperparameters.”**
>
> This is not correct. Our method trains a *linear* layer on frozen features. This optimization is extremely cheap due to its simplicity and convexity, taking literally seconds. It is also very insensitive to hyperparameters because of the convexity.
>
>
> > **”Please explain the meaning of '(almost) Provable”**
>
> “Almost” is used to highlight the fact that we need an assumption for the bound to hold provably. The error bound is guaranteed if the assumption is valid—but by definition, assumptions cannot be proven a priori, so it cannot be truly guaranteed in all settings. Notably, this is true of *all* methods that predict or bound error (including the baselines), even if they don't state it.
>
> Our extensive experiments across numerous benchmarks show that on natural distribution shifts the assumption holds and our method gives valid error bounds. Hence, though our method works well, “almost” is used to cover extreme scenarios where the assumption may not hold true, making the error bound incorrect.

---

> > ### Comment · Reviewer_PQBq · 2023-08-14
> > **Thank you for your replying!**
> >
> > Clarification on soundness 1: This poor score is mainly due to the evaluation metric and the unfair comparison.
> >
> > 1. The authors list several applications where safety is critical. However, the most reliable metric is the out-of-distribution (OOD) accuracy itself in these applications. As the proposed method requires touch to the unlabeled OOD data, the safest evaluation is to label the data and compute the out-of-distribution accuracy. If the authors want to demonstrate the necessity of the non-vacuous property of the proposed method, they should find an application that:  a. safety is essential, b. unlabeled OOD data is easy to be collected, c. labeling is expensive even considering the safety requirement.
> >
> > 2. Unfair comparison. It is not hard to make the other baseline methods non-vacuous. In practice, the simplest way is to add a safe threshold for prediction, which can be derived theoretically for every metric based on concentration inequality or expectation calculation by adding a $\sqrt{\frac{log \delta}{n}}$ term in the bound. This method would be similar to the shift setting provided in Appendix D when ignoring the dataset size difference. As shown in that table, some baseline methods can achieve around 90% coverage while the MAE is still smaller than the proposed method. In that sense, the results in Table 1 are misleading, which implies that it's hard for other methods to achieve a large coverage rate.
> >
> > Other discussion based on the reply from the authors:
> >
> > 1. Thank you for your clarification on your methods! However, the reviewer then has some questions on Assumption 3.5. As the authors only finetune the linear layer, Assumption 3.5 may not hold very well. Therefore, the reviewer wonders how much this term contributes to the error of the proposed bound. Although the authors provide a demonstration in Appendix E, the reviewer thinks that using the corollary to demonstrate the assumption is not straightforward. Additionally, the experimental results in the appendix are not positive: only 25.7% of experiments support the assumption. The reviewer wonders if this means that for 74.3% of experiments, the assumption is violated and the proposed bound is 'vacuous' in the perspective of theory.
> >
> > 2. To better demonstrate the assumption, the reviewer suggests the author to plot the relation between $\hat \Delta(\hat h, y^*)$ and $\hat \Delta(\hat h, h')$. The reviewer is aware that $\hat \Delta(\hat h, y^*) > \hat \Delta(\hat h, h')$ does not directly lead to $\Delta(\hat h, y*) > \Delta(\hat h, h')$, but such a plot can help the reviewer to identify if the assumption holds in a reasonable setting.

---

> > > ### Author Response · Authors · 2023-08-14
> > > **Responding to new items**
> > >
> > > Thanks for replying to our rebuttal!
> > >
> > > > “the safest evaluation is to label the data…find an application that: a. safety is essential, b. unlabeled OOD data is easy to be collected, c. labeling is expensive even considering the safety requirement.”
> > >
> > > We agree that these conditions (a, b, and c) are relevant. In fact, **we think it is hard to argue that the examples we gave in our original response, such as self-driving cars and medical diagnoses, do not already satisfy all of the above points.**
> > >
> > > **We use the case of medical diagnosis under hospital locale shift to show how a, b and c are satisfied.** (a) is satisfied as safety is essential in medical diagnosis applications; and (b) unlabeled patient data is easily available for a target hospital. Moreover, labeling medical data can be *very expensive* and needs expert human input. It’s also common for **the true diagnosis to be unknown,** or to see a shift **at test-time, when labeling data is impossible. So (c) is satisfied as well.**
> > >
> > > Finally, suppose safety on a particular task is important enough to warrant the labeling cost. **If our method guarantees high accuracy on one shift, we can avoid the cost of labeling it and instead label a different one.** This clearly shows the value of a *bound* rather than just an estimate. If a method is good on average but sometimes fails completely, is that really a method we should rely on?
> > >
> > > > “It is not hard to make the other baseline methods non-vacuous. [It] can be derived theoretically for every metric based on concentration inequality.”
> > >
> > > **We believe this claim is incorrect.**
> > >
> > > To clarify terminology, when you write “non-vacuous”, do you mean “guaranteed”? The baselines here are error *estimates*---as they do not bound the error like our method does, the term “non-vacuous” does not apply to them.
> > >
> > > Next, when you suggest this approach, **what quantity are you claiming will concentrate? Estimates by existing methods will not concentrate around the *true* test error; they will just concentrate around their *expected prediction*.** If this expected prediction is incorrect, no amount of data will cause them to give a valid bound. Thus, we do not believe simply adding a concentration term could allow prior methods to give true error bounds.
> > >
> > > **To clarify here, could you please state this claim in a more mathematically precise way?  We would be happy to discuss this further.**
> > >
> > > > “some baseline methods can achieve around 90% coverage while the MAE is still smaller than the proposed method.”
> > >
> > > There’s a key distinction here between *known, a priori bounds* and *post-hoc evaluation, reported for comparison*. Appendix D shows that other methods can get reasonable coverage (but not at the desired rate) *if we know exactly the correct shift/scale ahead of time.* Put another way, **these values represent test data leakage; they are only to show the "ideal" baseline (which we still often beat). It is not valid to cherry-pick the best performing item in the group.**
> > >
> > > If we try a baseline with fifty hyperparameter settings and a few of them do better, it would not be correct to say that that method is just as good—**in a *real* OOD setting, we would not know which setting to use, or whether the desired rate $\delta$ would be satisfied.** In contrast, our bound is valid every time without modifications.
> > >
> > > > “Although the authors provide a demonstration in Appendix E, the reviewer thinks that using the corollary to demonstrate the assumption is not straightforward”
> > >
> > > We are not sure we understand this statement. **Our corollary *proves* that the Assumption holds with very high probability on 25% of the datasets we evaluate on.** Could you clarify what you mean when you say this is “not straightforward”?
> > >
> > > > “The reviewer wonders if this means that for 74.3% of experiments, the assumption is violated and the proposed bound is 'vacuous' in the perspective of theory.”
> > >
> > > **We address this in detail in the paper. In the paragraph immediately after the one you are referencing (line 729), we wrote:** “Note that the fact that the bound is *not* violated for a given shift does not at all imply that the assumption is not true.”
> > >
> > > Also, **it seems like you may be using “vacuous” synonymously with “valid”. Please note that these are very different terms.**
> > >
> > > To clarify: A “valid” bound is one that is *correct*. A “vacuous” bound is one that is *trivial* (greater than the obvious bound of 1). A vacuous bound is always valid. The challenge is in giving *valid, **non**-vacuous* bounds, as our work does.
> > >
> > > > “the reviewer suggests the author to plot the relation between $\hat\Delta(\hat h, y^\*)$ and $\hat\Delta(\hat h, h’)$”
> > >
> > > **We *have* plotted this, it is in our response pdf. Our paper also already includes a similar plot, Fig. 4a.** In the last plot in our pdf, we plot “drop in accuracy” (i.e., $\hat\Delta(\hat h, y^\*)$) vs. “predicted drop in accuracy” (i.e., $\hat\Delta(\hat h, h’)$). **As you can see, the assumption holds consistently.**

---

> > > > ### Comment · Reviewer_PQBq · 2023-08-15
> > > > **Thank you for your correction**
> > > >
> > > > I want to point out that the authors keep avoiding to mention the role of term $\sqrt{\frac{\log 1/\delta}{n}}$ in both conventional bound and the proposed bound. In fact, for most previous OOD bounds, they usually predict an unbiased estimator $\hat f$ such that under some assumption, $Accuracy = \mathbb{E}\hat f$ and estimate OOD accuracy through $\frac{1}{n}\sum\hat f$. The author claims that those bounds can overestimate the target accuracy and the coverage is not good for those bounds. However, based on the concentration inequality, to keep those bound not overestimate, one can simply add the term $\sqrt{\frac{Var[\hat f]\log 1/\delta}{n}}$ to give an upper bound of accuracy which holds within probability $1-\delta$. This term, in principle, has no fundamental difference with the last term in Theorem 3.6. As shown in Fig4 (a), without this term, the proposed bound also overestimates the target accuracy. The authors warped this term for other bounds into a post evaluation through the so-called shift setting, but it indeed is a prior estimation. The relation between $\hat \Delta(\hat h, y^*)$ and $\hat Delta(\hat h, h')$ is a way I want to check the bound without the influence of such term and the source domain accuracy, but the author refuse to give such a result.
> > > >
> > > > By the way, the incorrect x,y label in the additional file can mislead the reviewer. I don't know why the authors do not fix the plot but add a notation on the caption, which seems like that they are aware of such a mistake and refuse to correct.

---

> > > > > ### Author Response · Authors · 2023-08-16
> > > > >
> > > > > Ah! Now we think we understand what you are saying, but **we are concerned that you may have misinterpreted the bound we derive in this work.**
> > > > >
> > > > > To start with,
> > > > >
> > > > > > “However, based on the concentration inequality, to keep those bound not overestimate, one can simply add the term to give an upper bound of accuracy which holds within probability $1 - \delta$.”
> > > > >
> > > > > ### **We don’t think this is correct—could you please make this a *precise* mathematical statement?**
> > > > >
> > > > > For discussion on why we think it's incorrect, see the main point next:
> > > > >
> > > > > ----
> > > > >
> > > > > > “As shown in Fig4 (a), without this term, the proposed bound also overestimates the target accuracy.”
> > > > >
> > > > > **Here is what we are interpreting your response here to be saying:**
> > > > >
> > > > > It sounds like you are claiming that when the concentration term is not included (as in Figure 4a), our method outputs a number that is smaller than the true error, and therefore it also “overestimates” the target accuracy.
> > > > >
> > > > > **Is this what you are saying?** Because if so,
> > > > > ### we wish to clarify that ***this interpretation of the experimental results is mistaken.*** To explain why in detail:
> > > > >
> > > > > The concentration term is included in our bound because **our assumption is on the *population* terms $\Delta, \epsilon_S$, but we can only evaluate the *empirical* terms $\hat \Delta, \hat\epsilon_S$**.
> > > > >
> > > > > First, note that if Assumption 3.5 holds, this immediately implies $\epsilon_T(\hat h) \leq \epsilon_S(\hat h) + \Delta(\hat h, h^\*)$ (note these are *population* quantities).
> > > > > ### Keep this in mind—Assumption 3.5 gives a valid upper bound, *if* we know the population quantities $\epsilon_S, \Delta$.
> > > > >
> > > > > However, **we cannot evaluate these quantities,** because we only have finite samples. Therefore, we must estimate them via their empirical counterparts. This is the source of the concentration term in our bound.
> > > > >
> > > > > Crucially, **this term does not concentrate to an *estimate* of test error**—it concentrates to *an error upper bound.* As we’ve pointed out, this approach would *not* work for other methods because
> > > > > ### concentration of $\hat f$ doesn’t help if the expectation $\mathbb{E}[\hat f]$ itself is not the true test error.
> > > > > You’ve introduced $\hat f$ as an “unbiased estimator” of the test error—but *we are unaware of any prior method which does this* **(certainly, none of the baselines we consider do this)**
> > > > >
> > > > > Instead, our term represents the concentration of $\hat \epsilon_T, \hat\Delta$ to their expectations, *which then gives a valid bound via Assumption 3.5 as we noted above.*
> > > > > ### This is the fundamental difference between our term (which gives a bound) and what you’ve suggested, which would *not* give a valid bound unless $\mathbb{E}[\hat f] = \epsilon_T$.
> > > > >
> > > > > Again, **we know of no such $\hat f$. If you have a *specific* one in mind, please share it with us.**
> > > > >
> > > > > Now that we’ve established the difference, it is important to observe that
> > > > > * **the fact that the value output by our bound without this concentration term is not an upper bound on the error does not mean that our bound is invalid or that Assumption 3.5 doesn’t hold.**
> > > > >
> > > > > Specifically, you wanted to see $\hat \Delta(\hat h, h^\*)$ vs. $\hat \Delta(\hat h, y^\*)$—but the fact is that
> > > > > ### it is **not possible** to draw conclusions about Assumption 3.5 by comparing these quantities, since Assumption 3.5 concerns the *population* quantities, which we do not know.
> > > > >
> > > > > The **only way** to know the assumption doesn’t hold is if our bound is invalid with the concentration term. If **(and *only* if)** our method gave an invalid bound *with* the concentration term,
> > > > > then we could conclude (whp) that Assumption 3.5 does not hold **(this is corollary E.1).**
> > > > >
> > > > > ### To reiterate: we cannot make any deductions from the empirical values alone because *Assumption 3.5 doesn’t say anything about the empirical values.*
> > > > >
> > > > > **We hope that this is now clear!** Please let us know if you have any other questions.
> > > > >
> > > > > ---
> > > > >
> > > > > > “the authors keep avoiding to mention the role of term in both conventional bound”
> > > > >
> > > > > We are not sure which “conventional bound” you are referring to.
> > > > > ### **Is there a specific reference that you could share with us?**
> > > > >
> > > > > > for most previous OOD bounds...predict an unbiased estimator $\hat f$ such that under some assumption, Accuracy = $\mathbb{E}[\hat f]$
> > > > >
> > > > > As we said above, we are not familiar with any method for bounding OOD error which uses an approach like this.
> > > > > ### **Could you please give us a few *specific* examples of bounds that do this?**
> > > > >
> > > > > > “The author claims that those bounds can overestimate the target accuracy”
> > > > >
> > > > > We would like to clarify here: **Are you discussing the baselines we compare to, which *estimate* accuracy, or work on *bounds***, which give vacuous results?
> > > > >
> > > > > We claim that prior work on *estimation* frequently overestimates test accuracy. As proof of this, see Figure 1. These methods do not provide bounds.
> > > > > ### **Which are you talking about here?**

---

> > > > > > ### Comment · Reviewer_PQBq · 2023-08-16
> > > > > >
> > > > > > Thank you for your reply. As the discussion before is so long that I am not sure if the authors are aware of what we are arguing about, I'd like to restate my main concerns and add some reply.
> > > > > >
> > > > > > 1. **Unfair evaluation**. The proposed method uses a concentration term to bound the outlier, but the authors do not incorporate this term into other methods. The authors have requested an example of the estimator, but I would like to point out that nearly all of the baseline methods in their table 1 have an expectation form. Thus there is no fundamental problem to check if those bounds can be valid for the target accuracy with the concentration term. It is the authors' responsibility to provide these results, rather than the reviewer's. And I believe their results in the appendix demonstrate that the baseline method can achieve a high coverage rate with a smaller MAE when employing the shift setting -- a modified interpretation of the concentration term. Consequently, it appears to me that the proposed method is not better than the baseline methods in practice, even under the authors' defined metric. However, in the main text, the authors only compare their method with baseline methods that lack a concentration bound term, which causes me to believe the evaluation result is weak.
> > > > > >
> > > > > > 2. **Assumption problem**. According to the authors, their bound can be theoretically proven, but as mentioned in the title and rebuttal, the bound is "almost" proven due to a strong assumption. Hence, verifying this assumption is a crucial task for the paper. However, the authors put this crucial data to the appendix. Furthermore, the assumption is only validated in 30% of settings, and for the remaining 70%, the assumption is not verified and thus one can claim that the proposed bound is not theoretically proven. Therefore, compared to other baseline methods, the proposed method has no theoretical advantage. Considering that the authors claim their method is applicable to safety-critical tasks, the insufficient verification results undermine the soundness of the paper.
> > > > > >
> > > > > > 3. **The practice usage of the metric**. The authors provide multiple safety applications and assert that they meet all my requirements. I disagree with this assertion as I believe no labeling task is more expensive than people's health. However, I will not debate this point as it is not within my area of expertise.
> > > > > >
> > > > > > Conclusion for now: During the rebuttal period, only one concern has been partially resolved. The authors have attempted to mask their unfair evaluation with theoretical results, but have not mentioned that the crucial assumption has only been verified on a 30% setting in the main text. Before the rebuttal period, I only found the evaluation problem in this paper. During the rebuttal, a clarification from the authors helped me find the assumption problem. It is unfair to reduce my score because the new problem is identified through the author's help. Thus, for now, I would keep my score.

---

> > > > > > > ### Author Response · Authors · 2023-08-17
> > > > > > >
> > > > > > > Thanks for stating these concerns clearly. In the interest of brevity, we’ll try to address each with a few sentences:
> > > > > > >
> > > > > > > > “The proposed method uses a concentration term to bound the outlier, but the authors do not incorporate this term into other methods.”
> > > > > > >
> > > > > > > We reproduced the baselines with the inclusion of the relevant concentration term. While some of the points are now below the line instead of above, *a large fraction are still above*, including the large overestimates in the top left corner of Figure 1. We believe this is sufficient to show that even with the concentration term, **these methods do not give valid error bounds, even empirically.**
> > > > > > >
> > > > > > > > “the proposed bound is not theoretically proven”
> > > > > > >
> > > > > > > **This is simply not true.** All theoretical results make assumptions; whether or not the assumption is true in a given instance does not affect the validity of the result. We also strongly disagree that this assumption is strong—we provide *multiple sources* [1,2,3] which suggest that it holds in many reasonable settings. Furthermore, we *prove* that it holds on a good fraction of the datasets, and **empirically, our bound is always valid** (on non-DA representations, and unlike all previous methods), which suggests that it likely holds for the remainder. Note that by definition it’s not possible to confirm an assumption in all settings, or it wouldn’t be an assumption.
> > > > > > >
> > > > > > > > “I disagree with this assertion as I believe no labeling task is more expensive than people's health.”
> > > > > > >
> > > > > > > We agree that in healthcare it is often worth the cost to attain labels. However, as we pointed out, it is sometimes not a question of cost—sometimes it is simply *not possible* to get labels, such as when the true diagnosis is not known. We are also not experts in healthcare, but **it sounds like we can all agree here that this is a subjective opinion on the motivation and not a flaw in the paper.**
> > > > > > >
> > > > > > > [1]  Last layer re-training is sufficient for robustness to spurious correlations. Kirichenko et al.
> > > > > > >
> > > > > > > [2] Domain-adjusted regression or: Erm may already learn features sufficient for out-of-distribution generalization. Rosenfeld et al.
> > > > > > >
> > > > > > > [3]  Decoupling representation and classifier for long-tailed recognition. Kang et al.

---

> > > > > > > > ### Comment · Reviewer_PQBq · 2023-08-18
> > > > > > > >
> > > > > > > > Thank you for you reply.
> > > > > > > >
> > > > > > > > 1. Thank you for your effort on producing the results. I'd like to see the accuracy distribution as Figure 1 and some metric evaluation as in Table 1. Where can I find those results?
> > > > > > > >
> > > > > > > > 2. Thank you for your response. However, the authors only cite a small part of my sentence. The original sentence should be ''Furthermore, the assumption is only validated in 30% of settings, and for the remaining 70%, the assumption is not verified and thus one can claim that the proposed bound is not theoretically proven''. It means that in a large part of settings (over 70%), the assumption has not been verified, so the validity of that bound is not guaranteed. I think such a statement is objective and correct.
> > > > > > > > And the authors only checked where we should believe the assumption held but did not check where it may be invalid. Therefore, I insist the authors provide a plot of $\hat \Delta(h,y^*)$ and $\hat \Delta(h,h')$. I emphasize that I'm aware of $\hat \Delta(h,y^*) > \hat \Delta(h,h')$ does not mean $\Delta(h,y^*) > \Delta(h,h')$, but if there is a non-negligible margin between those two terms, the probability of the assumption held is small.

---

> > > > > > > > > ### Author Response · Authors · 2023-08-18
> > > > > > > > >
> > > > > > > > > You are insisting on knowing the margin $\hat\Delta(h, y^\*) - \hat \Delta(h, h')$.
> > > > > > > > >
> > > > > > > > > **We'd like to remind you once more that this information is already in the paper.**
> > > > > > > > >
> > > > > > > > > If you revisit the definition of these terms,
> > > > > > > > > $$\hat\Delta(h, y^\*) - \hat \Delta(h, h') = \hat\epsilon_T(h) - \hat\epsilon_S(h) - \hat \Delta(h, h') = \hat\epsilon_T(h) - \left( \hat\epsilon_S(h) + \hat \Delta(h, h') \right)$$
> > > > > > > > >
> > > > > > > > > The first term in this expression is the x-axis of the third box in Figure 4a. The second term is the y-axis of that box.

---

> > > > > > > > > > ### Comment · Reviewer_PQBq · 2023-08-18
> > > > > > > > > >
> > > > > > > > > > Such a plot does not meet my requirement. The large range of the source domain and target domain accuracy hides the scale of $\Delta$. And also I can't found which parts of data in this plot supports the assumption 3.5 through Corollary E.2. Please provide the information according to my requirements, and highlight which part of results makes the assumption 3.5 verified.

---

### Official Review · Reviewer_uhaU · 2023-07-04

**Soundness:** 3 good
**Presentation:** 3 good
**Contribution:** 3 good
**Rating:** 6
**Confidence:** 4

**Summary:**

The paper proposes a method that under certain assumptions provides upper bounds on the accuracy under distribution shift when provided with unlabelled test data from the shift's target distribution.

**Strengths:**

The assumptions necessary to obtain the bounds are clearly stated and discussed.

The problem of estimating performance under distribution shift is important, and it is not clear whether meaningful bounds are possible. As such, the attempt of providing almost provable bounds in the way that it is done here could be a good contribution in this direction.

The experimental evaluations are quite extensive and reasonable. Particularly the careful examinations of when the assumptions are fulfilled are important and support these assumptions.

**Weaknesses:**

The method strongly depends on the function class in which the optimal critic is searched for, and potentially also on its approximation quality within that class. This means that when one allows a perfect critic to be found, which intuitively would look like it should be allowed since it definitely exists if the source and target distributions have smaller overlap than the source test error, then the obtained bounds are vacuous.
On the other hand, one can not know a priori whether a function class is large enough to contain a critic that satisfies Assumption 3.3 and thus provides a trustworthy guaranteed bound.
These issues are clearly stated and discussed, and it is shown that the necessary assumptions easily hold when considering linear functions on features for standard networks and benchmarks.
However, it is not clear how reliably the method could be transferred to new, potentially less regular datasets and/or models. The bounds after all rely on assumptions that can only be confirmed empirically after the true labels of the target distribution are known.

Compared to other methods, the proposed $Dis^2$ method makes estimates on the accuracy on the target distribution that are worse than simple baselines. This means that the issues of the provability of the bounds mentioned above are important weaknesses.

In l. 215, it is not clear that the logistic surrogate is still a valid approximation after it is combined with the disagreement logistic loss. The combination of the two losses should be discussed as the sum looks straightforward but has as far as I can see no intrinsic reason to be natural. There is balancing done by leaving out the $1/\log|Y|$ term, which seems very ad hoc. Other combinations like e.g. sum of squares or other powers of the individual losses would appear just as valid. Would it make sense (for certain function classes) to do a constraint optimization of the disagreement on the target data while fixing the decisions on the source (i.e. $\epsilon_S{\hat{h}, h') = 0$)?

**Questions:**

Why is Method [1] (Agreement-on-the-line) not compared to in the evaluations? This should be possible after it is fit on some other shifted distributions.

Certain assumptions on the distribution shift seem to be necessary, besides the discussed model-specific settings. For example, the bounds would not hold for a shift with similar images but switched class labels. Can such extreme cases be formalized and excluded?

**Limitations:**

Whether the necessary assumptions hold is extensively discussed and tested for standard distribution shifts. It is however not clear to what kinds of situations one can or cannot expect the method to generalize.

---

> ### Author Rebuttal · Authors · 2023-08-10
>
> Thanks for your feedback! We address your comments below.
>
> > **The bounds…rely on assumptions…confirmed empirically after the true labels…are known**
>
> You’re correct that we can’t confirm our assumption until the true labels are known. **It’s important to remember that this is true of *every* method, including the baselines.**
>
> The baselines also rely on assumptions—**their failures imply these assumptions are often not met.** The difference is that their assumptions are *unstated and unknown*, so it’s impossible in to even *guess* whether they will work. Instead, these papers use experiments to show their methods work—most of the time. But when they fail (which is not uncommon), **there’s no way to anticipate it.**
>
> We present experiments to demonstrate that our method is consistently reliable and accurate. But we also explicitly state a simple condition for guaranteed validity **and this allows us to identify failure cases a priori** (a very useful property).
>
> Since the focus of this paper is on *reliable bounds*, this represents a substantial improvement over prior work. Please see our overall response for further discussion.
>
> > **not clear how reliably the method transfers to new, potentially less regular datasets and/or models**
>
> As it’s impossible to evaluate on all datasets/models, **this statement is true for *every* ML method.** It is standard to evaluate on a wide set of benchmarks, with the hope that observations transfer.
>
> > **the proposed method…worse than baselines…provability of the bounds mentioned are important weaknesses**
>
> We agree that absent “true” provability, average accuracy is more important. But **a *true* guaranteed bound under shift is impossible without assumptions.** As we wrote above, the baselines we compare to all rely on *unknown* assumptions which could fail at any time. Further, all prior bounds are vacuous in practice.
>
> **The only way to be really, truly certain of a reliable error bound is to predict 100\% error.** We hope you agree that we should aspire to do better than that!
> This means that “reliability” is a continuum—in a given setting, we decide how *important* a valid bound is and choose the appropriate method.
>
> If “reliable bound” is the goal, **our method represents substantial progress.** Extensive experiments show $\text{DIS}^2$ consistently gives valid bounds. It does so while allowing us to identify failure cases a priori, at a small cost to overall accuracy. Furthermore, we can give relaxed bounds along the accuracy/reliability pareto frontier—**this is a very useful feature which other methods lack.**
>
> If reliability is not at all important, then it could make sense to use other methods; we tried to clearly convey this point several times throughout the paper. **But such a setting is not the focus of this work.**
>
> > **Why is Agreement-on-the-line not compared to?**
>
> We don’t compare to that method as the cost is substantially higher than just training the base model (it requires training many models for *each* distribution shift). In this work we consider methods with negligible overhead beyond training the model being evaluated. However, note that the pattern of severely underestimating test error is present in that method (e.g. the rightmost plots in Figure 1 of that paper).
>
> > **Certain assumptions on the distribution shift seem to be necessary, besides the discussed model-specific settings. For example…with similar images but switched class labels. Can such extreme cases be formalized and excluded?**
>
> We want to make sure we are understanding this point correctly. Are you saying that **(i)** this example implies assumptions on the distribution are necessary **in addition to** our existing assumption; or **(ii)** simply that one could make a more “fine-grained” assumption to exclude settings such as your example?
>
> If you mean **(i)**, then we want to clarify that **our bound does not need any additional conditions—Assumption 3.3 already handles this case.** Recall, Assumption 3.3 states that $\epsilon_T(\hat h, y^\* ) - \epsilon_S(\hat h, y^\*) \leq \epsilon_T(\hat h, h^\* ) - \epsilon_S(\hat h, h^\*)$ where $\hat h$ is the classifier we are evaluating, $y^\*$ is the true labeling function, and $h^\*$ is the critic that maximizes the disagreement discrepancy.
>
> In the setting you’ve described, there are two possibilities: either (a) the capacity of $\mathcal{H}$ is large enough to express a critic $h^\*$ which achieves larger discrepancy than $y^\*$, or (b) no such critic exists in $\mathcal{H}$. If (a), $\text{DIS}^2$ will predict test error close (perhaps equal) to 1. **This bound will be valid *and* very close to the true error.** If we are in setting (b), then the assumption is not met (note that (b) is exactly the negation of (a), and (a) is exactly Assumption 3.3).
>
> If you meant **(ii)**, note that our assumption *could* be made more mathematically precise as you are suggesting, but only by making additional distributional assumptions. Since our intent was to use the weakest assumption possible, we did not go this route.
>
> > **Logistic loss**
>
> Certainly, other combinations could work; since the precise loss was not our main focus, we didn’t extensively explore other options. The loss we derived works well—better than prior works. Designing alternatives could be an interesting follow-up.
>
> Leaving out the $\log Y$ term was intentional: since the features are optimized for “agreement” (the data is separable according to $\hat h$), equal weighting would give an unbalanced objective. We observed this in practice, so we rescaled.
>
> We did consider constraint optimization! Some algorithms have been designed for this in the fairness literature [1]. Unfortunately they only work for binary classification; we spent some time trying to extend it to multiclass but didn’t want to go too far down that path.
>
> [1] “Predictive Multiplicity in Classification.” Marx et al. 2020

---

> > ### Comment · Reviewer_uhaU · 2023-08-18
> > **Response to rebuttal**
> >
> > Thanks for the detailed responses to all points of my review!
> >
> > > We present experiments to demonstrate that our method is consistently reliable and accurate.
> >
> > From my understanding of when the provability of the bounds could be useful, these experimental results mostly confirm that the margin is large enough that underestimation is unlikely, but can as experiments not really confirm the provability.
> >
> > > But we also explicitly state a simple condition for guaranteed validity and this allows us to identify failure cases a priori (a very useful property).
> >
> > From my understanding, identifying the failure cases a priori relies on confirming the condition a priori, which is not possible. Do you mean something different here, or am I confusing something?
> >
> >
> > > transferability to new scenarios
> >
> > The reason I mentioned this point is that the method is supposed to give provable bounds. That the reliability of provable bounds when they are produced cannot be assumed is a drawback in my opinion that differs from other "no-free-lunch" situations, and for me strongly limits the meaning of provability.
> >
> > > The only way to be really, truly certain of a reliable error bound is to predict 100% error. We hope you agree that we should aspire to do better than that!
> >
> > I agree that such a bound would not be informative. However, I am not convinced that the bounds calculated in this paper do satisfy a useful notion of provable bound, since they rely on uncheckable assumptions in the moment the bound is computed and used. The point why I mention this as a weakness is that **if** one doesn't use the bounds, than the method is weaker than previous ones, which is an important point to me when evaluating the paper, even if the point doesn't contradict what is written in the paper which properly discusses it.
> >
> > > logistic loss
> >
> > My point here is that the justification from statistical learning theory for using the softmax convex surrogate makes sense if one optimizes it as the individual loss. Choosing a different surrogate for both parts still gives a convex function, but it might have a different mimimum than the sum of the original losses. The weighting question is basically almost the same point and can be (and maybe is) used to actively choose a combination with a minimum close to that of the original sum. However, while a theoretically precisely justified surrogate loss would be nice and strengthen the paper, I do not see this as a major weakness since the chosen loss works well enough empirically and could be improved in future works.
> >
> > ---
> >
> >  I still do not see the immediate the usefulness of guarantees that cannot be given without knowing measured results that could trivially be used to calculate even stronger guarantees / exact error estimations. However, while the assumptions cannot be verified a priori, as the authors state, these assumptions while unjudgeable in that sense are very weak and do not incorporate concrete knowledge on e.g. distributions. Thus I can see that there might be scenarios where they can be checked or be assumed with some concrete probability. This makes it plausible that the paper could be helpful for future works or situations where one has good prior reasons to assume that the assumptions hold. For the statements and methods that the paper provides, the technical and experimental treatment is good. I'm updating my current score from 4 to 6.

---

> > > ### Author Response · Authors · 2023-08-18
> > >
> > > **Thanks so much for taking the time to read our rebuttal and update your review!** We’re happy to further clarify the first two points:
> > >
> > > > “these experimental results mostly confirm that the margin is large enough that underestimation is unlikely, but can as experiments not really confirm the provability.”
> > >
> > > You are absolutely correct that these experiments cannot prove the provability. We hope we did not give you the impression that we are arguing this point :-)
> > >
> > > Our emphasis here was on the consistency of these results: whereas other methods frequently overestimate the test accuracy, our bound is (empirically) almost always valid. In the same way that the average accuracy of prior methods is taken as evidence that they will *probably* be reasonably accurate in the future, we feel this supports the idea that our method is reasonably likely to give valid bounds—though of course not guaranteed, hence “almost”. Also note that conditioned on overestimation, (i.e., if we penalize overconfidence but not underconfidence), our method does substantially outperform prior work.
> > >
> > > > “From my understanding, identifying the failure cases a priori relies on confirming the condition a priori, which is not possible. Do you mean something different here, or am I confusing something?”
> > >
> > > Sorry for the confusion. What we meant here is that because we are able to explicitly and simply state the necessary condition for success, we can often reason whether the condition is likely to hold in a given scenario, even without knowing the labels.
> > >
> > > As an example of this, we discuss in Section 3.1 how the use of a domain-adversarial (DA) representation could be expected to violate the assumption *a priori*, precisely because the regularization term in DA methods implicitly minimizes $\Delta(h, h^\*)$. Thus, even without seeing the data we would know not to expect a valid bound in this setting (though interestingly, our method does get much better MAE here).
> > >
> > > **Thanks again for your efforts in reviewing and staying involved during this discussion. Please let us know If you have any remaining questions.**

---

### Official Review · Reviewer_TEVz · 2023-07-04

**Soundness:** 3 good
**Presentation:** 4 excellent
**Contribution:** 4 excellent
**Rating:** 7
**Confidence:** 3

**Summary:**

The paper aims to propose a way to characterize the error bounds under distribution shift with provable error guarantees. Based on several assumptions, the authors show that the target error bound can be reasonably estimated by maximizing the agreement within the desired classifier on source distribution and maximizing the disagreement on the target distribution, i.e. $DIS^2$. The authors justified the validity of $DIS^2$ by comparing it to other existing methods both from theoretical and empirical perspectives, and showed that it successfully upper bounds the error under distribution shift. The authors also discussed to what extent the assumptions hold in practice with empirical findings.


**Strengths:**

The problem of justifying the validity of the trained models under distribution shift is important in modern machine learning. The prior work based on data dependent uniform convergence or assumptions of how distribution shifts are relatively vacuous in practice. The author proposes a method to characterize the error bound of the trained model on target distribution that is theoretically sound with assumptions almost empirically verifiable, making it a significant progress in this area.


**Weaknesses:**

Weaknesses:

1. Although empirical evidence shows that both Assumptions 3.3 and 3.5 are reasonable, the observations are still based on limited data. There could be cases when the assumptions fail to hold. The paper discussed a setting (domain-adversarial learning) where the proposed method $DIS^2$ may be invalid. It is still unclear to me when the assumptions do or do not hold in practice. From a theoretical perspective, is it related to the representativeness of the hypothesis class H? On the empirical side, although except for the adversarial learning case the target function is rarely the worst case, I still feel there could be a possibility. After all that is the whole point of worst-case analysis. Hence, this point of view needs to be supported by more substantive evidence, probably based on real data.

2. The representativeness of the hypothesis class is partly based on how many features the algorithm uses in training. There is a section talking about how the number of features affects the value of the error bound (the logits). It would be interesting to discuss these empirical findings together with hypothesis class representation.


**Questions:**

See weaknesses.

**Limitations:**

Yes, the limitations have been adequately addressed.

---

> ### Author Rebuttal · Authors · 2023-08-10
>
> Thanks for your thorough review! We appreciate that you have recommended acceptance, and that you consider the presentation and contribution of this work to be of high quality.
>
> > **“There could be cases when the assumptions fail to hold”**
>
> We agree that there are cases where the assumption might not hold. The best we can do in practice is to test our method in a wide variety of settings. We emphasize that **explicitly stating our assumption is strictly superior to not giving any guarantees at all.** Prior methods also make assumptions, implicitly. So, there could just as easily be settings where *their* assumptions fail to hold, but that would be harder to anticipate since *we don’t know what those assumptions are.*
>
> > **”is it related to the representativeness of the hypothesis class H?”**
>
>
> The validity of the assumption depends on the *interaction* between the capacity of $\mathcal{H}$, the predictions of $\hat h$, and the true labeling function $y^\*$. This cannot be made more mathematically precise without making additional distributional assumptions. Since our intent was to use the weakest assumption possible, we did not go this route. (Very) roughly, you can think of it as assuming that $\mathcal{H}$ contains a function “somewhat close” to $y^*$ (e.g., if $y^\* \in \mathcal{H}$ then Assumption 3.3 is immediately satisfied).
>
> > **”discuss these empirical findings together with hypothesis class representation”**
>
> We are not entirely sure we understand you here. Are you suggesting that we move the empirical results on reducing the number of features (i.e., Figures C.4/C.5 in the Appendix) to this section in the main body? Or add some additional discussion there? If you could please clarify we will be happy to flesh out the writing as necessary.
>
> > **”although…the target function is rarely the worst case, I still feel there could be a possibility. After all that is the whole point of worst-case analysis”**
>
>
> We totally agree. **But all prior methods give vacuous error bounds under shift.**
>
> Fundamentally, any guarantee is going to require some assumption. Since all prior assumptions are too weak to give non-vacuous bounds, we need to explore the correct way of strengthening them to make any progress. This work represents one such attempt.
>
> You wrote that we haven’t provided real evidence that the worst case doesn’t actually happen. We believe the fact that our method consistently works *is* data-based evidence for this claim. Further, as we remark in the footnote on page 5, **the “true” worst case would be precisely 0\% test accuracy.** As far as we are aware, this doesn’t happen in reality.
>
> Our extensive experiments imply that the assumption holds in practice, which is the most anyone could hope for. If someone doesn’t believe it will hold, that’s fine—but if they want to say anything meaningful at all, they’ll need to choose some *other* assumption to make.

---

> > ### Comment · Reviewer_TEVz · 2023-08-20
> > **Official Comment by Reviewer TEVz**
> >
> > Thank you very much for the detailed response. I agree that it is reasonable to make assumptions for proving better bounds and testify how realistic they are with empirical evidence.
> >
> > I still feel the term "somewhat close to $y^*$" is not very precise. It will be better if the authors could support this argument with specific metrics, say L_2 distance between different hypotheses.
> >
> > In my understanding, the empirical findings of how reducing the number of features affects the error bounds, is theoretically equal to how restricting the representativeness of the hypothesis class affects the error bounds. Say if you just output the top 10 of the PCs, you only output hypotheses from a restricted hypothesis class (by restricting it to a subspace spanned by these 10 PCs). Hence, it will be more stringent to ensure that there exists a function in $\mathcal{H}$ "close enough" to $y^*$. This is related to the hypothesis class representation analysis.

---

### Official Review · Reviewer_JSAn · 2023-07-05

**Soundness:** 3 good
**Presentation:** 3 good
**Contribution:** 3 good
**Rating:** 6
**Confidence:** 3

**Summary:**

This paper provides new error bound of distribution shift based on a notion called disagreement discrepancy. By assuming that the model class has enough expressiveness, it is theoretically proven that the error can be bounded by the worst-case disagreement discrepancy. Empirical investigation shows that such a method provides valid error bounds.

**Strengths:**

1.	The proposed method is novel and interesting. The paper does a great work in demonstrating why such a measure is better than $\mathcal{H}-$ and $\mathcal{H}\Delta \mathcal{H}$-divergence.

2.	This paper provides exhaustive experiments to test the performance of proposed measure.


**Weaknesses:**

1.	The writing of this paper can be further polished. For example, the authors can briefly introduce $\mathcal{H}-$ and $\mathcal{H}\Delta \mathcal{H}$-divergence in Section 3.

2.	According to the experiments, the proposed measure is not better than existing baselines.


**Questions:**

1.	In line 246, what does “the logits output by $\hat{h}$” mean?

2.	The method bounds the  disagreement discrepancy between h and true label by the worst-case disagreement discrepancy with respect to h. However, when the model class is highly expressive, how we can expect the worst-case bound to be tight?

3.	Does the method preserve the order of error? That is, if we have the error of model A is larger than model B, can we observe that the Dis^2 of model A is also larger than model B?


**Limitations:**

Please refer to “weakness”.

---

> ### Author Rebuttal · Authors · 2023-08-10
>
> Thanks for your feedback! To address your points:
>
> > **“briefly introduce $\mathcal{H}$- and $\mathcal{H}\Delta\mathcal{H}$-divergence”**
>
> We do roughly describe them in Section 3.1, but we will add a more detailed description with the additional space.
>
> You’ve stated the writing could use improvement—since this is just one example, **could you indicate anywhere else you believe the writing could be improved?** We would very much like to present this work clearly.
>
> > **”the proposed measure is not better than existing baselines.”**
>
> We acknowledge this point and we tried to clearly convey it several times throughout the paper. **But we do not believe that this constitutes a weakness of our method**. The goal of our approach is to give valid, non-vacuous error *bounds*, with raw accuracy being a secondary objective. We discuss this in more detail in our overall response to all reviewers.
>
> > **what does “the logits output by $\hat h$” mean?**
>
> Recall that $\hat h$ is the last layer of the network, which linearly transforms the features $\phi(x)$ into logits $h^\top \phi(x)$ to then produce a vector of probabilities via softmax. We are saying that instead of optimizing the critic on the features $\phi(x)$, we can also optimize directly on the logits $h^\top \phi(x)$ (if there are $k$ classes, the logit space will be $k$-dimensional).
>
> > **”how we can expect the worst-case bound to be tight?”**
>
> This is an important point and is key to understanding our bound. In fact, assuming the validity condition is met, **our method will give a bound that is exactly as tight as possible. If it gives a loose bound, it means it would would be *improper* to give a bound that is any tighter.**
>
> We use the term “improper” here to distinguish from “incorrect”. Any bound that is not an equality can of course be tightened while remaining correct. But in this setting, we have *no knowledge* of the distribution shift beyond unlabeled data. Thus, for a given critic which implies large test error, there is no justification for the conclusion that this critic is less likely to be correct than the network itself. And if it *were* the correct function, it would imply the reported error on the test distribution. Therefore, we cannot rule out the possibility that our network has this error, and so it is “improper” to output a bound that is any tighter.
>
> **This is precisely the idea conveyed by Figure 2(c).** Recall that we don’t have labels for the test set (red triangles). Therefore, **both $y^\* = \hat h$ and $y^\* = y_3^\*$ can be considered equally plausible,** because they both perfectly match the training data. Here, $\text{DIS}^2$ would return a bound of 0.5. In one possible universe, we have $y^\* = y_3^\*$, then $\hat h$ will have test error 0.5 and our bound will be exact. In another universe our bound may be loose. **But we cannot distinguish between these universes without labels for $\mathcal{T}$**, and therefore it would be “improper” to output a bound tighter than 0.5.
>
> We hope the above discussion is clear!
>
> > **”Does the method preserve the order of error?”**
>
> Great question! The answer can be deduced from Figure 1: “preserving error order” is equivalent to points further to the right on the X-axis also being further to the top on the Y-axis. We see that $\text{DIS}^2$ does approximately preserve order—there are a few points with a different rank than would be predicted (other methods exhibit the same), but overall we see the desired pattern.
>
> However, we also observe that Figure 1 omits an additional factor, namely source accuracy. So one thing we realize would be useful to report is whether $\text{DIS}^2$ and other methods preserve order in estimating the *drop in accuracy* from source to target. We plotted the result in our pdf response and it looks quite similar to Figure 1—all methods are reasonably order-preserving, with occasional outliers. Thanks for bringing this up!
>
> **We hope our response above (and our general response to all reviewers) answers your questions and that you will consider improving your recommendation. Please let us know if you have any additional concerns.**

---

> > ### Comment · Reviewer_JSAn · 2023-08-20
> >
> > Thanks for the clarifications. I will keep my score.

---

### Official Review · Reviewer_rQbE · 2023-07-08

**Soundness:** 4 excellent
**Presentation:** 4 excellent
**Contribution:** 3 good
**Rating:** 7
**Confidence:** 3

**Summary:**

This paper describes a novel method to estimate the error of a classifier under a shift of distribution. It relies on finding a worst case "critic" classifier which maximizes disagreement on some unlabeled target domain and hence bounds the disagreement with the true labeling function. The authors apply several necessary ticks to make the bound tractable and non-vacuous: (i) they introduce a new disagreement loss which can be efficiently maximized to find the critique and (ii) they only consider linear critics. While those two aspects make it possible that computed bounds are violated, in practice they show their bound is competitive in terms of mean absolute error while providing a better coverage than other methods.


**Strengths:**

Clarity: I found the paper well written and enjoyable to read. Presenting complex ideas in simple terms is one of the contribution of this work. Related works are cited adequately, I would only suggest adding [1] which seems also relevant.

Originality: While I find the insights and derivation not entirely novel---the derivation in [2] also relies on the same idea of bounding disagreement on the target domain with a worst-case critic while agreeing on the source domain---using those with the goal to provide an error bound on the target distribution is novel to me. Furthermore, the authors propose a novel disagreement loss.

Significance: The proposed approach is simple to implement yet competitive, I can see it being useful to the community.

References:

[1] Predicting Out-of-Distribution Error with the Projection Norm (https://arxiv.org/abs/2202.05834)

[2] Agree to Disagree: Diversity through Disagreement for Better Transferability (https://arxiv.org/abs/2202.04414)

**Weaknesses:**

While they are an extreme case, datasets with completely spurious correlations such as the ones in [3,4,2] might be interesting to consider. For those datasets, the best critic is actually $y^*$. You can then see if your approach recovers the right critic despite your added linearity constraint. As those datasets are shown in [4] to be a failure case when relying on logits, this might fail unless other features are used.

References:

[3] Evading the Simplicity Bias: Training a Diverse Set of Models Discovers Solutions with Superior OOD Generalization (https://arxiv.org/abs/2105.05612)

[4] Last Layer Re-Training is Sufficient for Robustness to Spurious Correlations (https://arxiv.org/abs/2204.02937)

**Questions:**

See weaknesses.

**Limitations:**

Limitations of the approach are mentioned adequately in various parts of the paper.

---

> ### Author Rebuttal · Authors · 2023-08-10
>
> Thanks so much for your thoughtful feedback! **We are very happy to hear that you appreciated the presentation; we took great pains to present this work as clearly as possible.**
>
> We will add the suggested reference [1] to the related work section. We remark that it would not make sense to include this work as a baseline—despite the title, our understanding is that that method does not actually predict error, but rather ranks models to enable model selection.
>
> Thanks for the additional datasets; these will certainly contribute a better understanding of the resulting critic. Note that the best critic is a function of the *classifier accuracy*, not the dataset itself. That is, unless the classifier has 0\% train error and 100\% test error, it is not the case that the best critic $h^* = y^*$. You are correct that with the linearity constraint we may not recover the best critic, but the advantage of our approach is that *we don’t need to recover $y^\*$*, which means that *we don’t need the true $y^\*$ to be linear to give a valid bound!*
>
> We agree that the derivation is not completely new, and we are careful to properly attribute prior work: we emphasize the connection to $\mathcal{H}\Delta\mathcal{H}$-divergence at several points. On the other hand, **we do feel that the use of this idea to give distribution-free non-vacuous error bounds on modern deep networks under distribution shift**—the first such result, that we are aware of—is quite noteworthy.
>
> **Thanks again for your helpful suggestions. We hope you will advocate for this work during the reviewer discussion.**

---

### Official Review · Reviewer_xsfT · 2023-07-12

**Soundness:** 3 good
**Presentation:** 3 good
**Contribution:** 2 fair
**Rating:** 4
**Confidence:** 3

**Summary:**

The authors propose a novel approach to estimate error bounds for deep neural networks under distributions shift. The proposed bounds remedy the limitations of exiting bounds that either provide vacuous bounds or underestimate the error for a big fraction of shifts while also often requiring access to test labels. In contrast, the bounds proposed by the authors only use unlabeled test data under a simple, intuitive assumption about the hypothesis class's ability to achieve small loss on the unlabeled train and test distributions. More specifically, their approach involves optimizing a disagreement discrepancy between two classifiers using a novel disagreement loss. The experimental results show that the obtained bounds are non-vacuous and comparable to existing competitive baselines.





**Strengths:**

- The presentation is clear and the paper is overall well-written. The notation is also introduced properly and make the theory easy to follow.
- The limitations of previous works are also clearly stated and provide a good motivation for this work.
- One of the main promises of this approach is allowing the user to interpolate between robustness and accuracy depending on the preferred level of risk tolerance.
- The approach is overall simple and leads to non-vacuous bounds without using test labels.

**Weaknesses:**

- The presentation of the results is the main drawback of this work for me! The choice of aggregating all the results over all datasets, shifts, and training methods is strange and does not allow an objective evaluation of the performance of the bounds and analysis of their drawbacks and limitations for the different scenarios. It is possible that the proposed bounds underperform severely on most of the datasets, shifts and training methods and perform exceptionally well on a subset of them. It is also concerning that even the appendix does not include the detailed evaluation for each dataset and shift pair separately.
- Additionally, we see from Table 1 that the only scenario where the proposed bounds outperform existing baselines is if the concentration term in Theorem 3.6 s dropped at the expense of coverage. Hence, a more detailed and transparent presentation of the results is crucial to verify the claims made by the authors.
- Including standard errors in Table 1 would also be crucial to verifying the validity of the results given the wide range of scenarios for which the performance was aggregated.

**Questions:**

- How does the calculation of the bound scale with the size of the dataset and model architecture in practice?
- In which cases would you expect your bound to be vacuous?

**Limitations:**

The authors identified a setting where their proposed bounds may be invalid. The above questions and feedback may help identify more limitations of this work to be explicitly discussed in the main text.

---

> ### Author Rebuttal · Authors · 2023-08-10
>
> Thanks for your feedback! You’ve stated that you are mostly concerned with missing experimental results, which we are happy to add. **We put preliminary results in our pdf response, which we'll expand on in the paper.**
>
> We want to briefly explain why these results were not originally included. You believe we may be misrepresenting test error *prediction*, but we emphasize that **the focus of this work is on giving valid, non-vacuous *bounds***, with accuracy being secondary. Our experimental results are meant to emphasize coverage, but also show that we get competitive MAE.
>
> > **“choice of aggregating all the results”**
>
> We agree that aggregating can obscure information. Our pdf response includes a rough version which stratifies by training method. We observe that **the pattern of accurate error estimation is retained across individual strata.** Preliminary results across datasets indicate that $\text{DIS}^2$ does better or worse when other methods do: the worst is $.2293 \pm .062$ (for ATC: $.1736 \pm .039$; difficult to draw conclusions with large std errors), and the best is $.0766 \pm .023$ (ATC: $.0758 \pm .018$).
>
> We originally aggregated the results because **stratifying presents a *less informative* evaluation of bound validity. Without aggregation, there isn’t enough statistical power to reject the null hypothesis that our bound is valid at the chosen confidence level.** The only way to demonstrate that our method *doesn’t* work is to show a statistically significant difference between the confidence level and the observed bound violation rate (Figure 4b). By stratifying, we would not have enough samples (i.e., shifts) in each group to distinguish between a valid/invalid bound.
>
> > **“it is possible to underperform on most methods and perform well on a subset”**
>
> The results in the pdf show that this is not the case. This also seems unlikely in general since MAE is bounded in $[0,1]$, so the influence of outliers is limited. Note that this could *not* occur for validity bounds: if our method only gave valid bounds for a small subset, the enforced level $\delta$ would be violated.
>
> We hope this distinction is clear and that our new results convince you that our method predicts error robustly. As you’ve indicated that this is your main concern and that you otherwise like the paper, **please let us know if there are additional results you think are missing and we will be sure to include them.**
>
> > **“outperform existing baselines”**
>
> We again point out that *pure accuracy* is not the focus of this work. We present a non-vacuous error bound under shift, and we are careful to never claim more. **We state as early as the abstract that our method does not do better than predictive baselines.** Where our method *does* outperform existing methods is coverage/reliability. We are unsure how this presentation could be made more transparent.
>
> As you noted, $\text{DIS}^2$ matches/outperforms baselines when dropping the concentration term. **This as a *strict improvement*---the other methods provide no coverage guarantees at all!** Dropping the concentration term puts all methods on equal footing for comparing raw accuracy.
>
> When reliability is important, we would retain the concentration term. **But now it no longer makes sense to compare $\text{DIS}^2$ to prior work purely on the basis of MAE, because our method provides guarantees and the other methods do not.** Instead we consider coverage, and our results show that $\text{DIS}^2$ gets excellent coverage, while retaining competitive accuracy.
>
> When coverage is not important at all, other methods may be preferable, as we point out in the paper. We even go so far as to *strengthen the baselines* by exploring ways to improve their coverage without reducing accuracy (Appendix D).
>
> > **including standard errors**
>
> We’ve recreated Table 1 with standard errors in our pdf. Since “coverage” is a binary variable and we already report $\hat p$, the only missing data was $n=90$, which is available in the Appendix. Stratified across training methods there is not much variation in MAE: the worst is $.1589\pm .022$, and the best is $.1360\pm.024$.
>
> > **Bound calculation scaling**
>
> Calculating the bound is trivial. It takes ~5-10s on one GPU for the largest datasets with $n_{\mathcal{S}} + n_{\mathcal{T}} \approx$ 62K, less than a second for the smaller ones—this could be sped up with stochastic optimization. **It does not scale with the model architecture** because we are optimizing over the frozen features of the network—the cost to extract these features is one pass over the dataset, exactly the same as the cost to evaluate accuracy. The optimization is similarly cheap because it is a convex objective optimized over linear predictors.
>
> > **”When to expect the bound to be vacuous?”**
>
> **This is an important question and crucial to understanding what makes our method so much stronger than prior bounds.**
>
> For simplicity, consider an interpolating network with 100\% train accuracy. By definition, our method will give a vacuous bound when there is a linear critic which fully agrees on train and fully disagrees on test. **This is exactly when we would *want* to output a vacuous bound.**
>
> Knowing nothing about the test distribution a priori, we cannot conclude that this critic is less likely to be correct than the network itself. And if it *were* the correct function, it would imply 100\% test error. Therefore, we cannot rule out the possibility that our network has 100\% error, and so it is “correct” to output a vacuous bound (assuming our focus is on reliability, as in this work). In other words, the bound will be vacuous precisely when it is “correct” to output a vacuous bound (conservatively). We hope this explanation is clear.
>
> **We are committed to presenting this work transparently, and we will add the metrics you’ve requested. We hope this will convince you to improve your recommendation. Please let us know if you have any additional concerns.**

---

> > ### Author Response · Authors · 2023-08-20
> > **Following up**
> >
> > Hi, as the discussion period nears the end we wanted to check in one more time to see if you are willing to reconsider your score in light of our response and the new results you asked for.
> >
> > You wrote that the presentation of the results was your main concern---we hope that the new info you requested such as stratified experimental results and standard errors helps to address that. Were there any other results that you felt would strengthen the paper? Even if we don't have time to add them now, we could always add them to the final version.
> >
> > Please let us know if you have any remaining questions and we will do our best to respond to them before the end of the discussion period!

---

### Author Rebuttal · Authors · 2023-08-10

Thanks to all reviewers for their helpful comments and suggestions!

We worked very hard to present this work as clearly as possible, so we are glad to hear that **reviewers xsfT, rQbE, TEVz, and uhaU all found the writing clear and easy to follow (even “enjoyable to read”!)**

A few reviewers pointed out that our method does not match existing baselines on Mean Absolute Error (MAE). We want to emphasize that **the primary focus of this work is on giving valid, non-vacuous error *bounds***, (that is the title of the paper, after all!) with accurate prediction being secondary, though still important.

We believe we were as upfront as possible about this in our writeup. Indeed, **as early as the abstract we state that we do not beat the baselines purely on MAE.** Where our method does outperform existing methods is reliability and conditional MAE, *and it does so by a huge margin.*

We think that in safety-critical settings where reliability is important, **a minor drop in overall accuracy is a small price to pay for substantially more reliable error bounds.** This is precisely what our method offers.

Consider a scenario where a model will be making crucial decisions: self-driving cars, medical diagnoses, etc. If we deploy this model expecting 80% accuracy and its actual accuracy is 20%, the result could be very costly. In this setting, a valid error bound is *essential* and accuracy of an error prediction only matters if we can trust it. Here the baselines do *very poorly* and our method substantially improves on SOTA—to our knowledge it is the first to give non-vacuous bounds. **If you agree that there are settings where average accuracy is secondary to reliability,** then we see no reason that having slightly worse accuracy should be considered a weakness, given that our method has substantially better coverage.

The main takeaway from our experimental results is that if we *only* care about accuracy, other methods might be preferable, as we state throughout the paper. **But if we are at all worried about overestimating test accuracy then coverage becomes important—and $\text{DIS}^2$ gets *much* better coverage, and has much lower conditional MAE.** We discuss this point further in Appendix D, where we even attempt to *strengthen the baselines*—but we find that our method is still best.

Our response pdf includes an updated Table 1 with standard errors and Figure 1 stratified by training method, as requested by reviewer **xsfT**. The other plot depicts estimated vs. true *drop in accuracy* (as opposed to test accuracy), as suggested by reviewer **JSAn**.

---

### Decision · Program_Chairs · 2023-09-21

**Decision:**

Accept (poster)

**Comment:**

This meta review is based on the reviews, the authors rebuttal and the discussions with the reviewers, discussions with the SAC, and ultimately my own judgement on the paper. There was a majority feeling that the paper contributes sound and interesting contributions. I feel this work deserves to be featured at NeurIPS and will attract interest from the community. I would like to personally invite the authors to carefully revise their manuscript to take into account the remarks and suggestions made by reviewers - in particular, making the goals of the paper more clear might help (this is a concern raised by one of the reviewers, who unfortunately didn't engage with your rebuttal - sorry about this). Congratulations!